



# Atmospheric ammonia (NH₃) over the Paris megacity: 9 years of total column observations from ground-based infrared remote sensing

Benoît Tournadre[1][*], Pascale Chelin[1], Mokhtar Ray[1], Juan Cuesta[1], Rebecca D. Kutzner[1], Xavier Landsheere[1], Audrey Fortems-Cheiney[1][**], Jean-Marie Flaud[1], Frank Hase[2], Thomas Blumenstock[2], Johannes Orphal[2], Camille Viatte[3], and Claude Camy-Peyret[4]

[1]Laboratoire Interuniversitaire des Systèmes Atmosphériques (LISA), UMR CNRS 7583, Université Paris-Est Créteil, Université de Paris, Institut Pierre Simon Laplace (IPSL), Créteil, France
[*]currently at: Centre for Observation, Impacts, Energy, Mines ParisTech, Sophia Antipolis, France
[**]currently at: Laboratoire des Sciences du Climat et de l'Environnement, UMR 8212, CEA/Orme des Merisiers, 91191 Gif sur Yvette, France
[2]Institut für Meteorologie und Klimaforschung (IMK), Karlsruher Institut für Technologie (KIT), Karlsruhe, Germany
[3]Laboratoire Atmosphères, Milieux, Observations Spatiales (LATMOS), UMR CNRS 8190, UPMC, Université Versailles St. Quentin, Institut Pierre Simon Laplace, Paris, France
[4]Institut Pierre Simon Laplace (IPSL), UPMC/UVSQ, Paris, France

*Correspondence to*: Pascale Chelin (pascale.chelin@lisa.u-pec.fr)

**Abstract.** In this paper, we present the first multi-year time series of atmospheric NH₃ ground-based measurements in the Paris region (Créteil, 48.79°N, 2.44°E, France) retrieved with the mid-resolution "Observations of the Atmosphere by Solar absorption Infrared Spectroscopy" (OASIS) ground-based Fourier Transform infrared solar observatory. Located in an urban region, OASIS has previously been used for monitoring air quality (tropospheric ozone and carbon monoxide), thanks to its specific column sensitivity across the whole troposphere including the planetary boundary layer. A total of 4920 measurements of atmospheric total columns of ammonia have been obtained from 2009 to 2017, with uncertainties ranging from 20% to 35%, and are compared with NH₃ concentrations derived from the Infrared Atmospheric Sounding Interferometer (IASI). OASIS ground-based measurements show significant interannual, and seasonal variabilities of atmospheric ammonia. NH₃ total columns over the Paris megacity (12 million people) vary seasonally by 2 orders of magnitude, from approximately $10^{15}$ molecules.cm$^{-2}$ in winter to $10^{17}$ molecules.cm$^{-2}$ for spring peaks, probably due to springtime spreading of fertilizers on surrounding croplands. Also, we observe a correlation of daily NH₃-OASIS total columns with daily PM2.5 in situ measurements from the closest Airparif surface station in springtime.



# 1 Introduction

Ammonia ($NH_3$) is a reactive and volatile chemical species present in the atmosphere as a trace gas. As the main alkaline atmospheric molecule (Behera et al., 2013), it plays a major role in the formation of secondary fine particulate matter of ammonium salts and thus in particle air pollution (Seinfeld and Pandis, 2006). $NH_3$ is heterogeneously distributed in the

atmosphere, depending on local and regional sources (potentially including agriculture, urban traffic, particle-to-gas conversion, sewage, industrial activities) and sinks (wet and dry deposition, gas-to-particle conversion).

In France, atmospheric ammonia ($NH_3$) is mainly emitted (more than 94%) by agricultural activities (Ringuet et al., 2016, Genermont et al., 2018) via livestock waste and urine, manure spreading and the use of synthetic nitrogen fertilizers. Other minor sources exist, including the combustion of biomass and fossil fuels as well as industrial activity. The source related to

motor traffic has increased significantly in recent years, due to the increasing use of catalytic (or non-catalytic) $NO_x$ reduction systems on light and heavy duty vehicles. These devices use an injection of urea or ammonia and can give rise to $NH_3$ emissions (Chang et al., 2016). The urban area corresponding to the Paris city and its suburbs is the second largest megacity in Europe with more than 12 million people. Ammonia emissions in the Paris region reached nearly 11 kilotons in 2014 of which 93% are attributed to agricultural activities, with a negligible contribution of livestock (DRIEE, 2017). The

Paris megacity is located at the administrative region Île-de-France, which includes 49% of agricultural land, with particularly strong activity in the eastern half of the territory. Île-de-France agriculture is mainly dedicated to crops, in particular cereals (with a dominance of wheat), which occupy 67% of the cultivated area. This sector is described as an agricultural belt, with 59% in the Seine-et-Marne department. Urban traffic in the Paris region is estimated to be responsible for 5% of ammonia emissions and industry for 2% (DRIEE, 2017).

Atmospheric ammonia concentrations vary to a large degree during the day, since the atmospheric lifetime of $NH_3$ is rather short, on the order of hours to a few days (Galloway et al., 2003). Wet and dry depositions dominate the atmospheric sink of this inorganic compound. $NH_3$ is also a gaseous precursor of fine particulate matter (PM2.5, particles of an aerodynamic diameter less than 2.5 μm). As the main alkaline molecule in the atmosphere, it reacts rapidly with sulfuric ($H_2SO_4$) and nitric ($HNO_3$) acids in the atmosphere to form ammonium sulfate or ammonium nitrate (Behera et al., 2013). These

ammonium salts may represent more than 50% of the PM2.5 fraction, during peaks of spring air pollution, periods of fertilizer application by agriculture. Concentrations of ammonium nitrate in Île-de-France in the 2009/2010 period can be simulated by the CHIMERE chemistry-transport model with only a small bias (<20%, Petetin et al., 2016). However, comparisons of the precursors' concentrations measured during the FRANCIPOL campaign (May 2010 to February 2011) show an overestimation of nitric acid and an underestimation of ammonia by the CHIMERE model (Petetin et al., 2016).

These differences appear to stem in part from the uncertainties associated with ammonia emissions. They suggest large uncertainties in the emissions of the precursors of ammonium nitrate over the Paris region.

Monitoring the atmospheric concentrations of $NH_3$ is essential for improving air quality models, as well as for quantifying the fluxes characterizing the nitrogen cycle. The measurement of atmospheric ammonia concentrations is not subject to



regulation (in the sense of the European Directives) and is really challenging for several reasons: (i) High temporal and spatial dependence of ambient concentration levels (Allen et al. al., 1988; Sutton et al., 1998); (ii) Rapid conversion of $NH_3$ between gaseous, particulate and aqueous phases (Warneck, 1988); (iii) Measurement artefacts due to $NH_3$ reactivity with sampling and measurement systems: $NH_4^+$-$NH_3$ conversion, adsorption on surfaces, etc. (Sutton et al., 2008; Von Bobrutzki

et al., 2010). This is why there are only a few measuring stations for which atmospheric ammonia concentrations are measured with a time resolution finer than a month. Most of the existing observation networks are designed to evaluate long-term trends and are therefore based on sampling techniques accumulating during several weeks (passive samplers, denuders or liquid bubblers) and off-line analysis in the laboratory.

Other in situ methods enable to estimate surface atmospheric concentrations of $NH_3$ (e.g. by cavity ring-down spectroscopy

(CRDS) techniques (Sun et al., 2015)) or even in the lower atmosphere (airborne campaigns (Leen et al., 2013, Shephard et al. al., 2015, Sun et al., 2015)). Hourly measurements are very rare at present, and are mainly available during intensive short-term measurement campaigns and are limited to a small number of measurement sites.

An innovative and very promising alternative for monitoring atmospheric ammonia is infrared remote sensing. This approach exploits the ammonia absorption spectral signatures of thermal infrared radiation measured by Fourier transform

infrared spectroscopy (FTIR) instruments. These methods are free from sampling problems and are notably less influenced by local sources than in situ observations. This is currently, or until very recently, performed from space using measurements of the IASI sounder (Infrared Atmospheric Sounding Interferometer, onboard the Metop satellites, Clerbaux et al, 2009; Clarisse et al., 2009), TES (Tropospheric Emission Spectrometer embedded on the Aura satellite, Shephard et al., 2015, TES's mission was ended in January 2018, after nearly a 14-year career of discovery), CrIS (Cross-track Infrared Sounder

embedded in the Suomi National Polar-Orbiting Partnership satellite, Dammers et al., 2017) and AIRS (Atmospheric Infrared Sounder, Warner et al., 2016, here, a grating spectroscopy instrument installed on the Aqua satellite). These measurements allow the retrieval of total columns of $NH_3$, vertically integrated concentrations between the Earth's surface and the top of the atmosphere. In particular for the IASI instrument, observation processing algorithms have evolved considerably in recent years (Van Damme et al 2014, 2015; Whitburn et al. 2016), a data product that we call hereafter $NH_3$-

IASI. These currently available satellite data also feature indicators of the quality of the space-based observations and retrievals, and have been validated (Dammers et al., 2016). Even though the observations from space remain complex due to the weak absorption of this species, the availability of a global distribution of $NH_3$ twice a day (for example over Europe) is an important achievement. However, the precision of these satellite measurements is about 50% at most (Van Damme et al., 2014) because of small absorption ammonia features recording at a coarse spectral resolution, the sensitivity to surface

concentrations of $NH_3$-IASI is also limited, and the IASI morning and evening overpasses are not ideal for measuring $NH_3$ peaks.

In addition, the high-resolution FTIR solar stations of the Network for the Detection of Atmospheric Composition Change (NDACC) measure, from the ground, the total ammonia columns but are also capable of providing information on its vertical distribution (Dammers et al., 2015). Spectroscopic measurements from the ground have been used for decades in the



validation of satellite measurements. The diurnal variability is observed directly from the ground, with a temporal resolution of a few minutes. The high temporal resolution of these ground-based solar measurements can help to understand the nature of ammonia sources, especially in urban environments where there is a crucial lack of observations of this short-lived and spatially highly disperse pollutant. A better knowledge of $NH_3$ emissions and associated atmospheric processes thus requires
extended observational networks able to assess high temporal and spatial variabilities of the atmospheric content of ammonia. Such networks are still to be built, especially in France where $NH_3$ measurements are extremely rare.

In this study, we present the first multi-year time series of atmospheric $NH_3$ ground-based measurements over a European megacity (Paris), retrieved with the moderate-cost mid-resolution OASIS (Observations of the Atmosphere by Solar absorption Infrared Spectroscopy) FTIR solar observatory located at Paris suburbs (Créteil, France). Given its good
sensitivity to surface pollutant concentrations, it has been used previously for monitoring urban pollution (tropospheric ozone and carbon monoxide) (Viatte et al., 2011; Chelin et al., 2014). A total of 4920 measurements of $NH_3$ total columns have been obtained between 2009 and 2017, with uncertainties ranging from 20% to 35% (one standard deviation) using the retrieval code PROFFIT (Hase et al., 2004) adapted for our medium spectral resolution and based on the NDACC stations' methodology. The data are compared with $NH_3$ concentrations from IASI measurements for verifying consistency with the
$NH_3$-OASIS retrievals.

The paper is organized as follows. First, we present in detail the $NH_3$-OASIS data by describing the site, the retrieval strategy and uncertainties (Section 2). A comparison between the $NH_3$-OASIS columns and the $NH_3$-IASI satellite data is provided in Section 3.1, while Section 3.2 gives the atmospheric $NH_3$ time-series focusing on seasonal variability. Finally, in Section 3.3, we examine the correlation between atmospheric ammonia measured by OASIS and surface PM2.5
concentrations at a daily scale.

## 2 Ground-based FTIR $NH_3$ data: description and characterization

### 2.1 The FTIR-OASIS observatory

The OASIS ("Observations of the Atmosphere by Solar Infrared Spectroscopy", 48.79° N, 2.44° E, 56 m above sea level) observatory is located in the Paris region, which is a European megacity (12 million inhabitants) surrounded by a large rural
region and relatively flat terrain (Figure 1). It routinely performs solar absorption measurements since 2009 under clear-sky conditions, using a mid-spectral resolution spectrometer (BRUKER Vertex 80, with a spectral resolution of 0.06 cm$^{-1}$, maximum optical path difference of 12 cm). In order to carry out air quality research, we have assessed the capability of a medium-resolution FTIR solar absorption spectrometer for monitoring pollutants, especially $O_3$ and CO. We have demonstrated that OASIS is able to continuously monitor tropospheric ozone over Créteil with good accuracy and sufficient
information content (Viatte et al., 2011). A 5-year analysis of ozone in the lower troposphere and carbon monoxide has also been made (Chelin et al., 2014). Given the moderate cost and compactness of OASIS, deployment of analogous systems nearby or in large megacities might be useful in support of satellite and air quality studies in other regions of the world.



The observatory comprises an automatized cupola (Sirius 3.5 "School Model" observatory, 3.25 m high and 3.5 m in diameter) in which the upper part (a dome equipped with a mobile aperture) rotates to track the sun. The alt-azimuthal solar tracker in OASIS is the A547N model manufactured by Bruker Optics, using bare gold-coated mirrors which are less sensitive to corrosion and pollution than the original Al mirrors. Infrared solar absorption spectra are nominally recorded on

a deuterated-triglycine-sulfate (DTGS) detector using a potassium bromide (KBr) beamsplitter, in order to cover the spectral region from 700 to 11000 $cm^{-1}$ (0.9 to 14.3 μm) without any optical filter, so that column abundances of many different atmospheric trace gases can be retrieved simultaneously.

## 2.2 Ground-based $NH_3$ retrievals

The ammonia spectral signatures used in this study are observed in the 10.6 μm spectral region and belong to the $\nu_2$

vibrational band (Dhib et al., 2007). To achieve a sufficiently high signal-to-noise ratio, each spectrum is produced by co-adding 30 scans at the highest spectral resolution, resulting in one interferogram recorded over a period of approximately 10 min. Each coadded interferogram is Fourier-transformed for obtaining a spectrum without further numerical apodization (i.e. unapodized/boxcar apodization). Carbonyl sulfide (OCS) cell measurements are regularly performed to verify the alignment of the instrument (Chelin et al., 2014). Ammonia absorption lines from the $\nu_2$ vibrational band are also used for satellite-

based estimates (Clarisse et al., 2009; Whitburn et al., 2016) and in the first retrievals of ammonia from high-resolution ground-based NDACC FTIR stations (Dammers et al., 2015). Atmospheric transmission spectra from two spectral microwindows are used in this work (926.3-933.9 $cm^{-1}$ and 962.5-970 $cm^{-1}$). They are slightly larger than those for the Bremen and Lauder NDACC stations (Dammers et al., 2015) because of the coarser spectral resolution of OASIS atmospheric spectra (0.06 $cm^{-1}$). The main interfering species in these windows are $H_2O$, $CO_2$, $O_3$, which are simultaneously

retrieved together with $NH_3$. Minor interfering species are $HNO_3$, $SF_6$, $C_2H_4$, and CFC-12. Figure 2 shows an example of a measured spectrum on the 21st March 2012 with contributions of all main species, here calculated using the spectral atlas of Meier at al. (Meier et al., 2004). The strong spectral signatures of ammonia are seen in both microwindows, even when observed with the medium spectral resolution of the Vertex 80 spectrometer. Note that the $NH_3$ concentrations of this day were particularly higher than on average, resulting in very strong $NH_3$ features.

The retrievals are performed using the PROFFIT 9.6 code (Hase et al., 2004) widely used by the NDACC community to retrieve trace gases from high-resolution FTIR measurements, but adapted for the medium resolution. The software is capable of including spectral channeling in the fitting process and for the estimation of error budgets. Channeling is caused by the presence of optical resonators as e.g. filters or windows, in the measurement beam. Here, the ability of the code to handle spectral channeling in the fit is of relevance, because we intend to quantify a minor absorber and some channeling

was detected in the measured spectra. For this purpose, the channeling frequency is determined from a Fourier transform of the residual of an auxiliary fit. A neighbouring spectral section containing few absorption lines and wide enough to encompass many cycles of the channeling signal is selected for this purpose. For the chosen frequency, a sine and a cosine amplitude are subsequently included in the fit of the $NH_3$ target windows. Daily temperature and pressure profiles for the





meteorological variables are obtained from the Goddard Space Flight Center NCEP (National Center for Environmental Prediction). For radiative transfer calculations, profiles at 44 altitude levels, from 50 m up to 70 km, are set. Spectroscopic data are taken from the HITRAN 2008 database (Rothman et al., 2009), except for $CO_2$ lines from the HITRAN 2012 database (Rothman et al., 2013). As the radiance values are rather small below 1000 cm$^{-1}$, a quality criterion was introduced

selecting only spectra with a signal-to-noise ratio higher than 30, as measured between 960 and 990 cm$^{-1}$ covering lines of a weak $CO_2$ band. Figure 3 shows a measured spectrum (black line), the corresponding simulated spectrum (red line) and the difference between observation and simulation (blue line) in both microwindows. The fits are excellent with a standard deviation of 2% in both microwindows.

**2.3 Degrees of freedom and uncertainty**

As for satellite retrievals and in consistency with the moderate spectral resolution of OASIS, we derive total columns of $NH_3$ from each radiance spectrum. A scaling factor of a climatological vertical profile of $NH_3$ is adjusted in order to minimize the difference between measured and simulated spectra. So that, the degree of freedom concerning ammonia retrievals is 1. The climatological a priori profile of $NH_3$ is taken from the MIPAS project (Remedios et al., 2007). The a priori profiles of the

interfering species ($H_2O$, $CO_2$, $O_3$) are taken from the Whole Atmosphere Community Climate Model (WACCM version 6) (Chang et al., 2008). We consider the posteriori errors calculated by PROFFIT 9.6. This calculation is based on the error estimation method by Rodgers (2000). For the uncertainty in the $NH_3$ line parameters, we assume values as stated in the HITRAN 2008 database (Rothman et al., 2009). We assume a conservative value of 20% for the integrated intensities and of 10% for the pressure broadening coefficients. The total errors are dominated by uncertainties in the spectroscopic

parameters, and are equivalent to those estimated by Dammers et al. (2015) for a high resolution ground-based station at Bremen (Germany). Table 1 summarizes the results of total errors according to the different ammonia total columns, which vary from 20% to 35%. Another a priori was tested with higher concentrations of $NH_3$ in the atmospheric boundary layer and has shown the same variabilities with relative differences contained in the error bars (only slightly higher ammonia total columns have been retrieved using this a priori profile: around +20% relative difference).

**3 Results**

**3.1 Comparison between NH$_3$-OASIS and NH$_3$-IASI**

The validation of the satellite products of IASI (Van Damme et al., 2014) is limited by the scarcity of long-term series of atmospheric ammonia measurements. A first attempt to validate IASI-$NH_3$ measurements was made with correlative data from surface in-situ and airplane-based measurements (Van Damme et al., 2015). They confirmed consistency between the

$NH_3$-IASI dataset and the available in-situ observations and showed promising results for validation by using independent airborne data from the CalNex campaign (California Research at the Nexus of Air Quality and Climate Change). Nevertheless, that study was limited by the availability of independent measurements and suffered from representativeness





issues for the satellite observations when comparing to surface concentration measurements. Recently, Dammers et al. (2016) reported a first step in the validation of $NH_3$-IASI products, comparing ammonia columns with high-resolution FTIR measurements from several NDACC stations around the world. They concluded that IASI reflects similar pollution levels and seasonal cycles as shown by FTIR observations and the best correlation (R = 0.83 and a slope of 0.60) was obtained with

the NDACC Bremen station. Bremen is located in the northwest of Germany, which is characterized by intensive agriculture. It is most suitable for comparisons with IASI given the very high atmospheric concentrations of $NH_3$ observed there. Compared to this work with ground-based measurements over Bremen, our analysis is the first data series over a megacity, Paris, during 9 years. First, we compare the $NH_3$-OASIS columns with $NH_3$-IASI data. In this study, we consider $NH_3$ total columns retrieved from the IASI-A instrument (aboard the Metop-A platform) observations along with those retrieved from

the IASI-B instrument (aboard the Metop-B platform) observations, IASI-B data being available only for the period from 8 March 2013. Only morning overpass (AM) satellite observations are considered here, as they are generally more sensitive to $NH_3$ owing to more favorable thermal contrast at daytime (Van Damme et al., 2014) and for better temporal coincidence (OASIS observations are only carried out during daytime). The satellite data have a circular footprint of 12 km diameter at nadir and an ellipsoid shaped footprint of up to 20 km×39 km at the outermost angles (Clarisse et al., 2009). We use raw

observations from 1 January 2009 to 31 December 2016, from the most recent version of IASI $NH_3$ retrievals (the near-real-time neural network retrieval version 2 with reanalysed meteorological inputs called ANNI-NH3-v2.2R, Van Damme et al., 2017).

Note that, the lifetime of atmospheric $NH_3$ is rather short, on the order of hours not being uncommon (Galloway et al., 2003), up to a few days, due to efficient deposition and fast conversion to particulate matter. Thus, $NH_3$ concentrations vary

strongly as function of emission strengths and meteorological conditions (such as temperature, precipitation, wind, vertical mixing in the atmospheric boundary layer).

In order to minimize differences associated with the temporal variability of $NH_3$, we only consider $NH_3$ measurements from OASIS performed within +/- 30 min with respect to the IASI morning overpass. The spatial coincidence criterion is 15 km between the center of IASI pixels and OASIS. We also exclude $NH_3$-IASI data with relative errors higher than 100% or with

absolute errors higher than $5.10^{15}$ molecules cm$^{-2}$ (only if relative errors are higher than 100%).

Figure 4 shows a scatterplot comparison between $NH_3$-OASIS and $NH_3$-IASI data (within 30 min interval and 15 km distance). We obtain a very good correlation of $R = 0.79$ and a slope of 0.73 for a total of 52 coincidences. These results are very similar to those from a comparison between a ground-based FTIR at Bremen and $NH_3$-IASI: R = 0.83, and a slope of

0.60 for a total of 53 coincidences (Dammers et al., 2016). Here for Paris, we used stricter spatiotemporal collocation criteria for the comparison, as we have more data than the Bremen study. For this correlation, we calculated the absolute differences (AD) between satellite (y axis) and FTIR-OASIS $NH_3$ total columns (x axis), which are defined here as:

$$\text{AD} = (NH_3\text{-}IASI) \; column - (NH_3\text{-}OASIS) \; column \quad (1)$$





The average of the absolute differences is -0.78 $10^{15}$ molecules $cm^{-2}$, with a root mean squared error (RMSE) equal to 4.86 $10^{15}$ molecules $cm^{-2}$ and a standard deviation of error (STDE) equal to 4.84 $10^{15}$ molecules $cm^{-2}$. This reveals a very good consistency between $NH_3$-IASI and $NH_3$-OASIS, this last one being analyzed during 9 years (2009-2017) in section 3.2.

We further investigate the representativeness of the OASIS site for the Île-de-France region by comparing $NH_3$-OASIS data with that from IASI at different distances from the ground-based site. We use 15-km-wide rings (Figure 5) centered on OASIS observatory and increasing the minimal distance ($d_{min}$) of the rings from OASIS observatory with a 1-km step. The 15-km width of the rings is a compromise to be the least affected by ammonia spatial variability but to maximize the number of coincidences for statistics. The minimal distance ($d_{min}$) varies from 0 km up to 400 km. Figures 6 and 7 show respectively

the correlation and regression slope as a function of $d_{min}$ using a maximum allowed sampling time difference of 30 min. The numbers on the right axis of each of the figures show the amount of coincident observations used in the comparison. An increasing $d_{min}$ shows a decreasing correlation (blue lines) and a changing slope (increasing with distance up to 120 km, then decreasing). One can distinguish three different regimes in the plot of Fig. 6. For $d_{min}$ between 0 and 12 km, highest correlations are seen with $R$ varying from 0.84 down to 0.57, then between 12 km and 120 km, $R$ is around 0.6 despite lots of

noise and then we get decreasing correlations for $d_{min}$ between 120 km and 400 km. This comparison underlines that measurements from OASIS observatory provide information about atmospheric $NH_3$ variability on a regional scale, up to 120 km away from the site, but might also be affected by more local processes and/or emissions.

For $d_{min}$ between 0 and 12 km, regression slopes vary between 0.73 and 0.81 revealing an underestimation of observed columns by IASI compared to OASIS (Fig. 7), already mentioned by Dammers et al. (2016) when evaluating IASI with

some high resolution NDACC stations. This underestimation may be explained by IASI's lower sensitivity to surface ammonia concentrations, due to the weak thermal contrast between the surface and lowermost troposphere.

## 3.2 Seasonal variability of $NH_3$ ground-based data

Figure 8 shows the multi-year time series of $NH_3$ total columns derived from FTIR ground-based OASIS measurements

retrieved from all the 4920 available spectra in 2009–2017, resulting from 234 measurement days. Table 2 gives a summary of statistics of the retrieved $NH_3$ measurements versus the four seasons. Individual measurements with an overall mean total column of 0.84 $10^{16}$ molecules $NH_3$ $cm^{-2}$ and a standard deviation of 0.86 $10^{16}$ molecules $NH_3$ $cm^{-2}$ indicate a large variability in the observations. They highlight peak abundances in spring (March-April-May), more precisely in March. The amplitude of the spring peaks varies throughout the years, with maxima in March 2012, reaching about 9 $10^{16}$ molecules

$NH_3$ $cm^{-2}$. The occurrence of the highest $NH_3$ concentrations in March is particularly noticeable: all measurements are above 2 $10^{16}$ molecules $NH_3$ $cm^{-2}$, which corresponds to the mean of data plus one standard deviation over the springtime period (March/April/May). They are measured for this calendar month, for almost every year (2011 (2 days), 2012 (11 days), 2014 (6 days), 2015 (4 days), 2016 (4 days) and 2017 (1 day)). March 2012 is therefore a month particularly polluted in terms of





atmospheric $NH_3$. In addition to many days with more than $2 \ 10^{16}$ molecules $NH_3 \ cm^{-2}$, a peak reaching the quadruple of that threshold on March 21, 2012, represents the maximum retrieved ammonia total column.

The seasonal variability is analysed in Figure 9 in terms of monthly averages over the 2009-2017 period. The mean $NH_3$ column in March is $1.65 \ 10^{16}$ molecules $NH_3 \ cm^{-2}$, which is two times higher than the overall mean total column over the 4920 measurements in 9 years.

As for many other regions, $NH_3$ seasonality is well marked, with high values that might be connected with the timing of agricultural manure spreading. As shown by both approaches of Ramanantenasoa et al. (2018) for 2005-2006, mineral fertilizers are mainly used in the Île-de-France region because they are major arable crop (especially cereals) farming areas. They account for 59% of $NH_3$ emissions according to CADASTRE_NH3 framework based on the process-based Volt'Air model (Garcia et al., 2011; Garcia et al., 2012). These fertilizers are mostly spread during springtime. In Bremen, $NH_3$ atmospheric total columns have a similar seasonal cycle with highest levels during spring. The maximum values occur around April, which is consistent with temporal emission patterns for manure application reported for this region (Dammers et al., 2015). Note that for example, highest levels in Bremen were observed during springtime with total columns reaching up to $9.3 \ 10^{16}$ molecules $NH_3 \ cm^{-2}$, which is close to the maximum peak observed by OASIS in March 2012.

One can observe also in Figure 8 significant concentrations in June (2017), July (2010, 2012, 2013, 2014, 2015 and 2017) and in August 2016, with another threshold of $1.5 \ 10^{16}$ molecules $NH_3 \ cm^{-2}$, which corresponds roughly to the mean of data plus one standard deviation over the summer time period (June/July/August). This shows a second seasonal peak for the summer period. On the contrary, during winter months (December-January-February) the $NH_3$ total columns have a pronounced minimum (mean total column of $0.12 \ 10^{16}$ molecules $NH_3 \ cm^{-2}$ and a standard deviation of $0.12 \ 10^{16}$ molecules $NH_3 \ cm^{-2}$). It is worth noting that fewer observations are available during this season due to frequent overcast conditions.

These winter and, to a lesser extent, summer variabilities are also observed in the Bremen measurements, emphasizing some similarities between both the Paris and Bremen station environments. This seasonal behavior is not only found in Europe but also in other megacities as Seoul (Korea), where surface ammonia concentration exhibit higher values during the warm seasons, summer and spring, while dropping to a minimum in the cold season (winter) (Phan et al., 2013). One possible explanation is the increased volatility of ammonia (present in agricultural soils and atmosphere in aqueous or solid phase) with high temperatures. $NH_3$ volatilization and its influence of fine particles such as ammonium salts is beyond the scope of this paper and will be discussed in detail in our next study on the diurnal analysis of total and surface ammonia measurements from Paris region during a high pollution event. Hence we need data with high temporal resolution that can be obtained from the OASIS ground-based infrared remote sensing observatory.



### 3.3 Comparison of daily NH₃-OASIS and in-situ PM2.5

As mentioned, the highest ammonia total columns in the period 2009-2017 over the Paris region were measured in March 2012. Since ammonia is a precursor of PM2.5, as shown in Fortems-Cheiney et al. (2016) during a European spring haze episode, we expect a link between high ammonia concentrations and inorganic salts, such as ammonium nitrate. That period

during late 2012 winter (documented by Petit et al. (2014)), was probably the most polluted March of the last ten years in Paris region (Petit et al., 2017). We further investigate the correlation between daily $NH_3$ total columns of averaged daytime measurements from OASIS and surface PM2.5 concentrations from the Airparif network ([https://www.airparif.asso.fr)](https://www.airparif.asso.fr)) that hourly monitors air pollutants such as ozone, carbon monoxide, nitrogen oxides, PM, over Île-de-France. For the comparison with OASIS measurements, we selected the nearest Airparif station, located at about 5 km from the observatory, in Vitry-

sur-Seine, and defined as background station.

For each year in 2009-2017, we calculated daily PM2.5 concentrations over springtime (March-May), which exceed the World Health Organization (WHO) target daily value of 25 $\mu g.m^{-3}$ (WHO, 2005). We find maximum number of daily exceedances in March 2012 (20 days), compared to 12 days in March 2014 and March 2015 that are well known PM springtime pollution episodes over France (Petit et al., 2017, Tarrasón et al., 2016).

In different studies (Bressi et al, 2013, Petit et al., 2014 ; Petit et al., 2015), measurements of inorganic species (nitrate, sulfate, ammonium and chloride) over Île-de-France region were conducted at the SIRTA supersite over Île-de-France (Site Instrumental de Recherche par Télédétection Atmosphérique, 2.15° E; 48.71°N; 150 m above sea level.; Haeffelin et al., 2005; [http://sirta.ipsl.fr](http://sirta.ipsl.fr)). These measurements show that about 50% of PM2.5 or PM1 (submicron particles) during spring over Paris is composed of ammonium nitrate (Bressi et al., 2013; Petit et al., 2014; 2015).

Figure 10 shows the correlation as a function of the season between $NH_3$-OASIS and surface PM2.5. The highest correlation is found during springtime. In this case, the correlation coefficient is 0.63, showing a noticeable positive covariation between daily $NH_3$ total columns and daily surface PM2.5. This is probably associated with the concomitant occurrence of polluted days in terms of $NH_3$ and PM. For the other seasons, correlations are lower. Moreover, when comparing hourly values of $NH_3$ and PM2.5 (without daily average) for springtime, correlation decreases down to 0.47 (603 hourly coincidences),

showing that the diurnal evolution of total ammonia columns and PM2.5 need to be further investigated.

### 4 Conclusions and perspectives

Ground-based infrared remote-sensing is undoubtedly a promising and powerful spectroscopic technique to retrieve ammonia columns, even using FTIR instruments with moderate spectral resolution such as OASIS. Recording one spectrum over a period of approximately 10 min, the OASIS instrument can provide over 40 spectra per day over the Paris region and





allows the observations of the diurnal cycle of ammonia. In this paper, we presented the first multi-year time series (2009-2017) of atmospheric $NH_3$ total column measurements from ground-based infrared remote-sensing over the Paris megacity. $NH_3$ total columns vary seasonally by 2 orders of magnitude, approximately from $10^{15}$ molecules.cm$^{-2}$ in winter to $10^{17}$ molecules.cm$^{-2}$ for spring peaks. Error estimations show random errors of about 10% and systematic errors less than 25% for

individual observations, mainly due to uncertainties in spectroscopic parameters.

Satellite remote-sensing instruments such as Infrared Atmospheric Sounding Interferometer (IASI) on board the Metop platforms provide global distributions of atmospheric $NH_3$ relying on valuable information (Van Damme et al., 2018). We have compared $NH_3$-IASI data with our measurements from OASIS over the 2009-2016 time period. We show a very good correlation of 0.79 and a mean bias of -0.78 $10^{15}$ molecules cm$^{-2}$ between the two datasets. This underestimation of the FTIR

columns by IASI may be explained by the space instrument's lower sensitivity to surface ammonia concentrations, due to the weak thermal contrast between the surface and lowermost troposphere.

Furthermore, OASIS observatory measurements can provide information on $NH_3$ time variability at a 100-km regional scale. Based on the 9-year time series, the study focuses on seasonal variability of atmospheric $NH_3$ in the Paris region and we found peaks especially in March. The predominance of $NH_3$ peaks occurring in March is particularly noticeable: all

measurements above 2 $10^{16}$ molecules $NH_3$ cm$^{-2}$, which corresponds to the mean of data plus one standard deviation over the springtime period (March/April/May), come from this month, and are well connected to the manure spreading time periods (Ramanantenasoa et al., 2018). Mineral fertilizers are mainly applied in Île-de-France region because they are major arable crop (especially cereals) farming areas and could result in high ammonia concentrations under sunny conditions when the solar OASIS measurements can only be performed. This study highlights also a marked seasonality for the summer period,

with another threshold of 1.5 $10^{16}$ molecules $NH_3$ cm$^{-2}$, which corresponds roughly to the mean of data plus one standard deviation over the June/July/August time period and pointing out a possible increased volatility of ammonia under warm meteorological conditions.

In a first step for analyzing the correlation between $NH_3$ and PM2.5, we observe that the daily $NH_3$ total columns measured by the OASIS ground-based observatory and the daily surface PM2.5 measured from the Airparif network (Vitry-sur-Seine

station) show a noticeable correlation in spring time of R ~ 0.63 and lower correlations for the other three seasons. On the contrary, if we focus on the hourly time series, correlations drop down even for spring time, highlighting a need to further investigate the link between diurnal evolution of total ammonia columns and fine particles. Ground based stations like OASIS are of major importance for validating current (IASI) and future satellite observations (Infrared Atmospheric Sounder Interferometer - New Generation (IASI-NG) and Meteosat Third Generation-InfraRed Sounder (MTG-IRS) of $NH_3$

in the infrared for a better understanding of the space and time variability of this major source of nitrogen species in the troposphere.



**Acknowledgments:**

The authors from LISA acknowledge support from University of Paris-Est Créteil and from the OSU-EFLUVE (Observatoire des Sciences de l'Univers-Enveloppes Fluides de la Ville à l'Exobiologie) to make the observatory still operational. IASI is a joint mission of EUMETSAT and the Centre National d'Etudes Spatiales (CNES, France). They

5    acknowledge the support of the Centre National des Etudes Spatiales (CNES) via the project IASI-TOSCA (Terre-Ocean-Surface-Continentale-Atmosphère). The authors also acknowledge the AERIS data infrastructure for providing access to the IASI data in this study and ULB-LATMOS for the development of the retrieval algorithms. The authors wish to thank Airparif for providing in-situ PM2.5 data, and the NASA Goddard Space Flight Center for providing the temperature and pressure profiles from the National Center for Environmental Prediction (NCEP). The research was also funded by DIM Qi2

10   (Region Île-de-France) for internship financial support and by LEFE-CHAT. Work at IMK is funded by the ATMO program of the Helmholtz Association of Germany Research Centres.



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



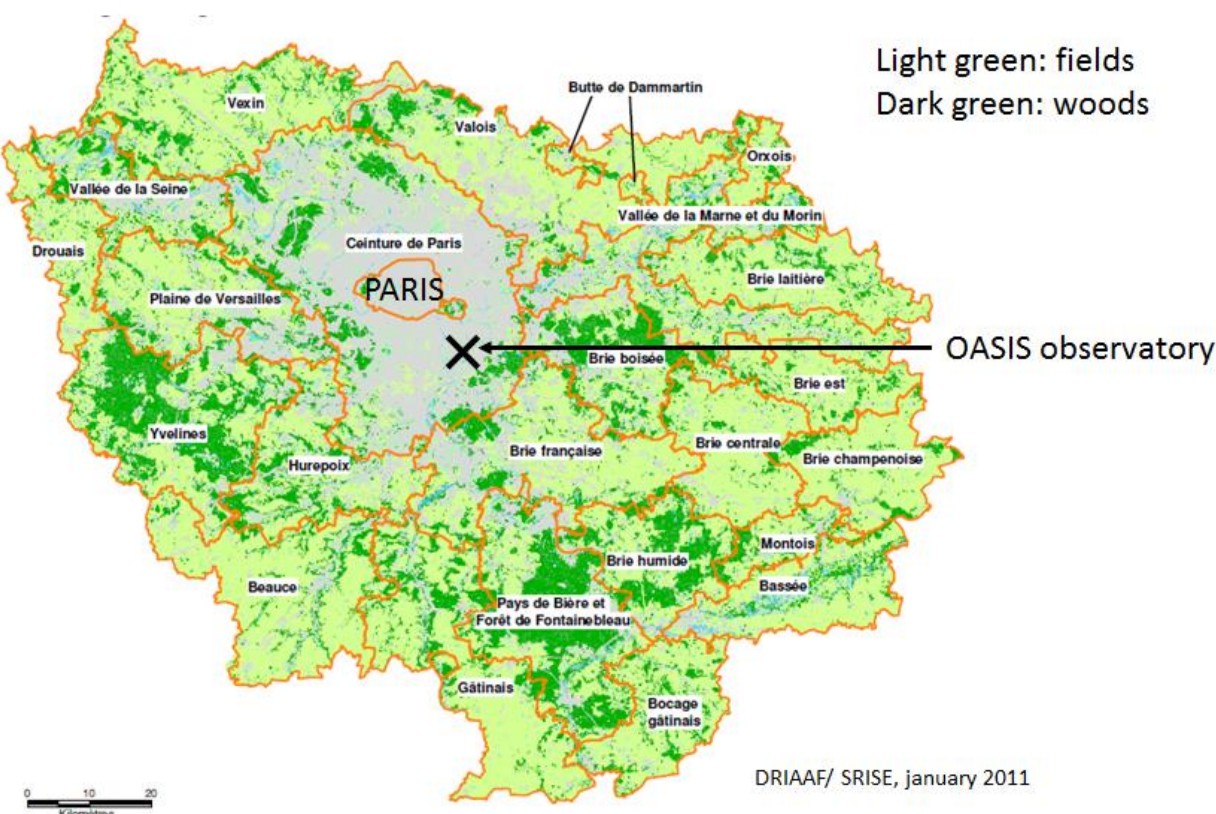

**Figure 1: Location of the FTIR OASIS observatory providing NH$_3$ total columns in Paris region (France). One can see that Paris megacity is very urbanized close to the Paris city, and surrounded by a large rural belt with 49% agriculture surfaces (source Agreste Île-de-France http://agreste.agriculture.gouv.fr/IMG/pdf/R1118C01.pdf).**



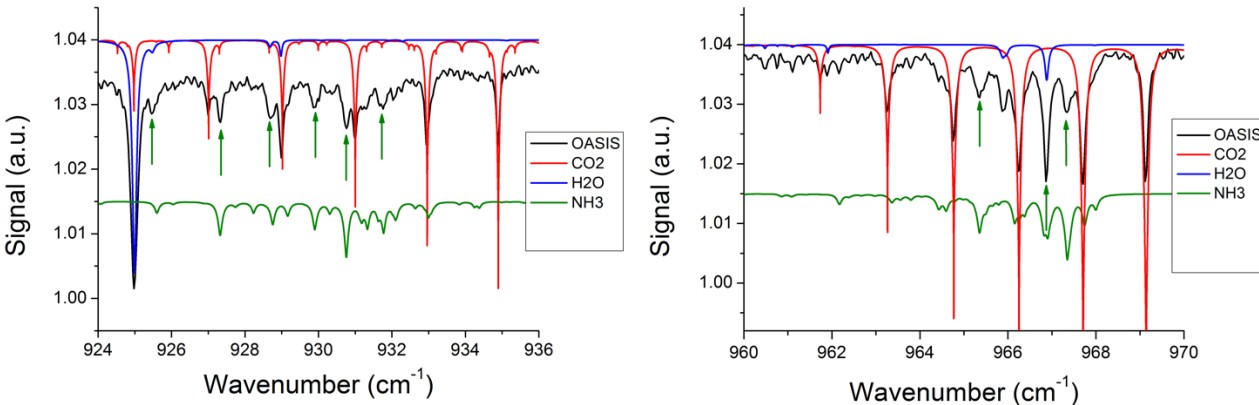

**Figure 2: Measured spectrum for both spectral windows obtained with the BRUKER Vertex 80 at Creteil on 21st March 2012, with individual contributions of the main absorbing species represented from the atlas of Meier et al. (Meier et al., 2004) in the first (on the left) and second (on the right) spectral windows.**





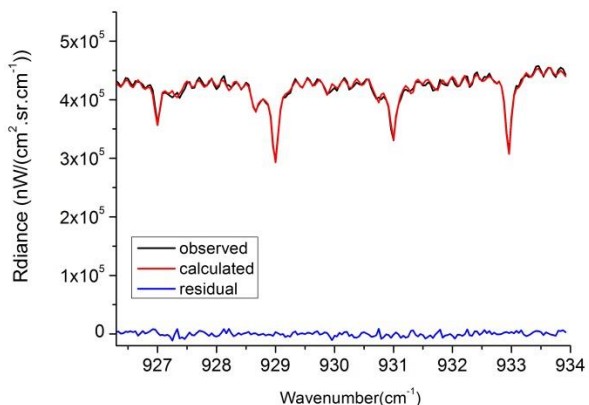
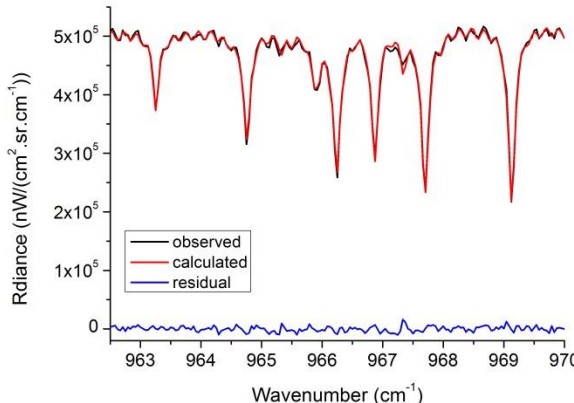

**Figure 3: Observed (black line) and calculated spectrum (red line) in radiance unit (nW/(cm².sr.cm⁻¹)) for both spectral windows measured with the BRUKER Vertex 80 in Creteil on the 21st March 2012 with the fitting residuals (blue line).**



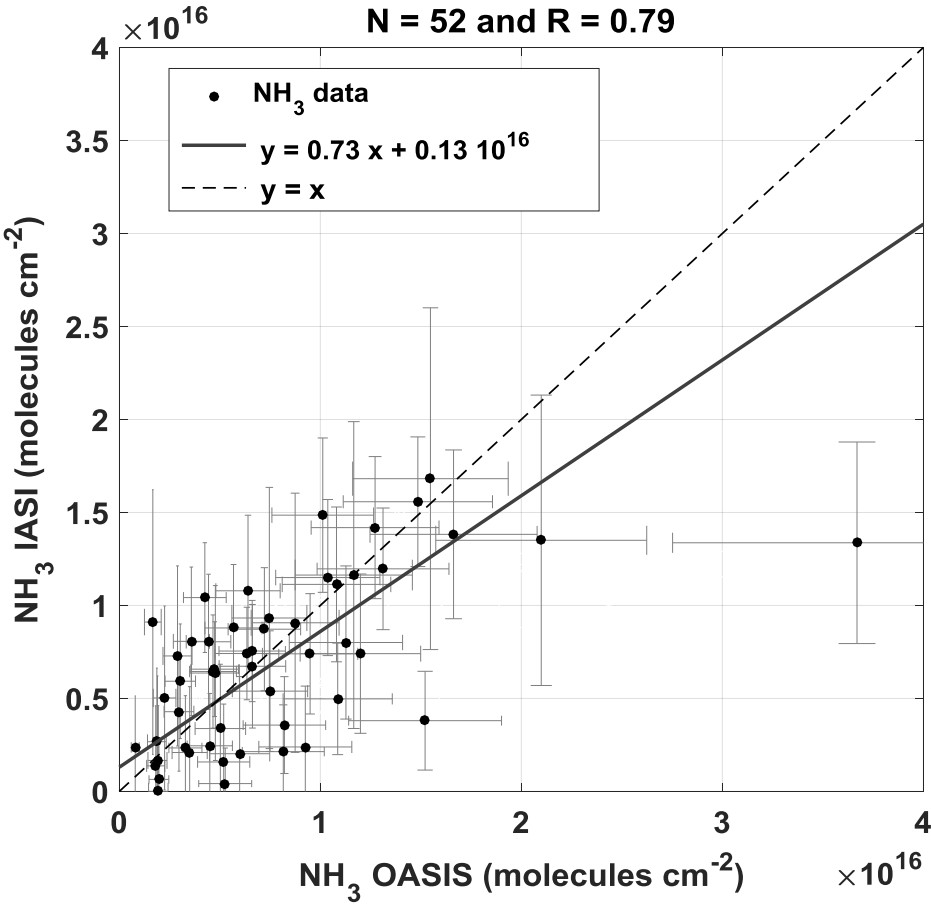

**Figure 4**: **Correlations between the FTIR/OASIS and IASI NH$_3$ total columns fulfilling the temporal and spatial coincidence criteria: a time difference smaller than 30 min, and a 15 km radius centered on the ground-based FTIR station. The dashed line is the straight equation with slope 1.**





$d_{max} - d_{min} = 15$ km

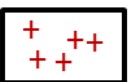 IASI pixels used to calculate
correlation $r$ with FTIR versus $d_{min}$

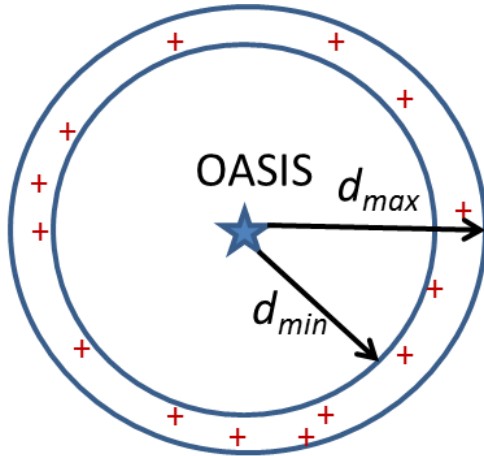

**Figure 5: Description of the 15-km rings centred on OASIS observatory that were used to select different IASI pixels for the correlation with FTIR data.**

**Figure 6: Correlation coefficients R (blue lines, left axis) between IASI and ground-based FTIR observations with temporal sampling difference smaller than 30 min, as a function of *dmin* to OASIS observatory, considered for the calculation of the 15 km wide rings. The data (orange lines, right axis) show the total number of IASI/OASIS coincident observations.**





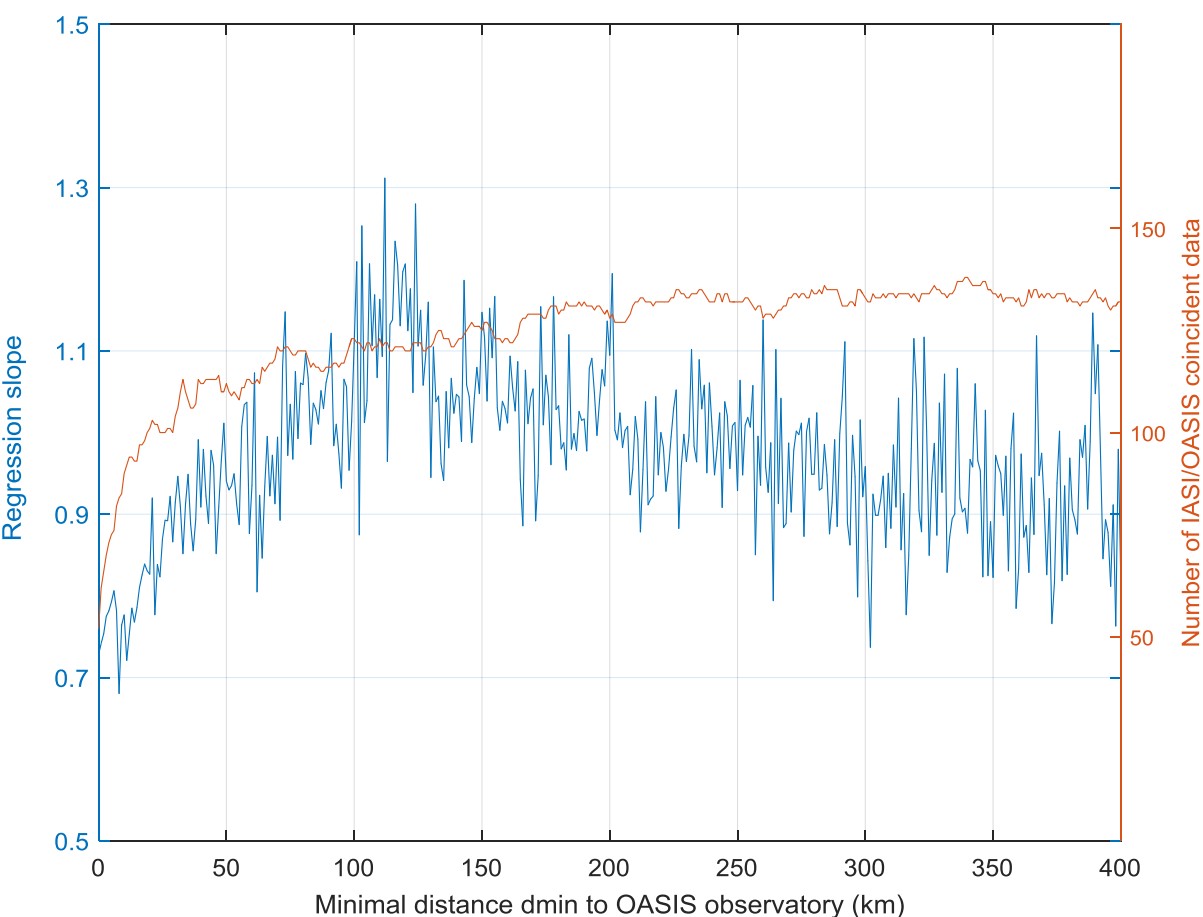

**Figure 7: Regression slope (blue lines, left axis) between IASI and ground-based FTIR observations with temporal sampling difference smaller than 30 min, as a function of *dmin* to OASIS observatory, considered for the calculation of the 15 km wide rings. The data (orange lines, right axis) show the total number of IASI/OASIS coincident observations.**

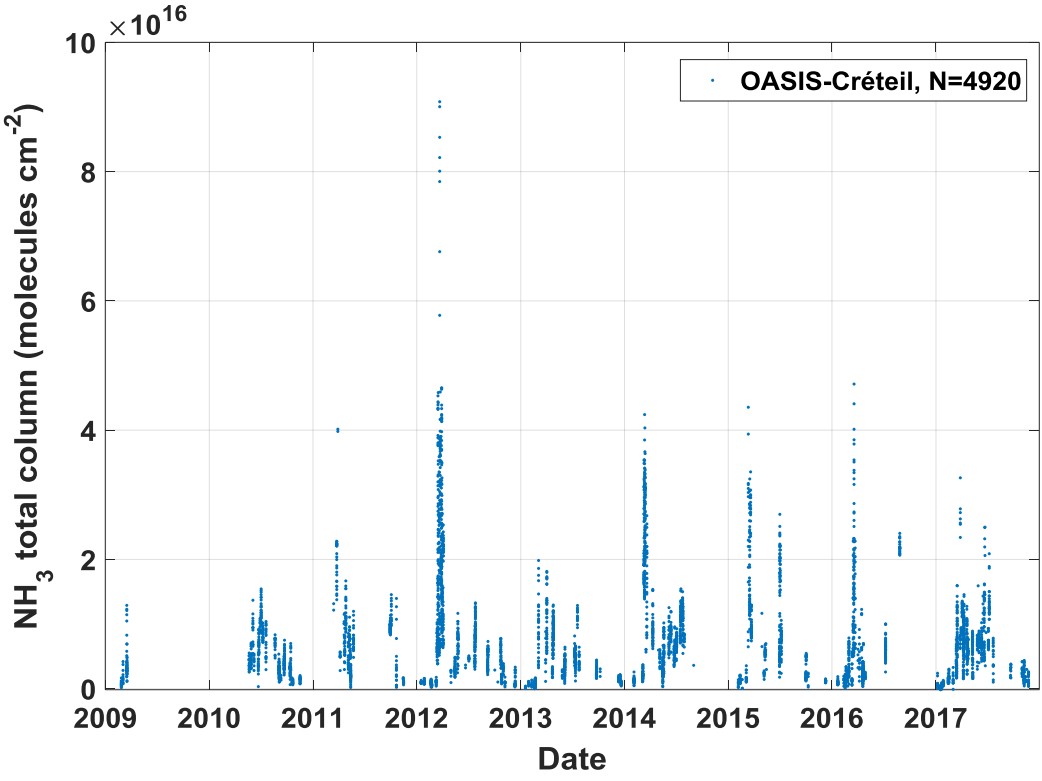

**Figure 8: First multi-year time series of NH$_3$ total columns derived from OASIS measurements over Paris, retrieved from 4920 infrared atmospheric transmission spectra measured during 234 days between 2009 and 2017.**




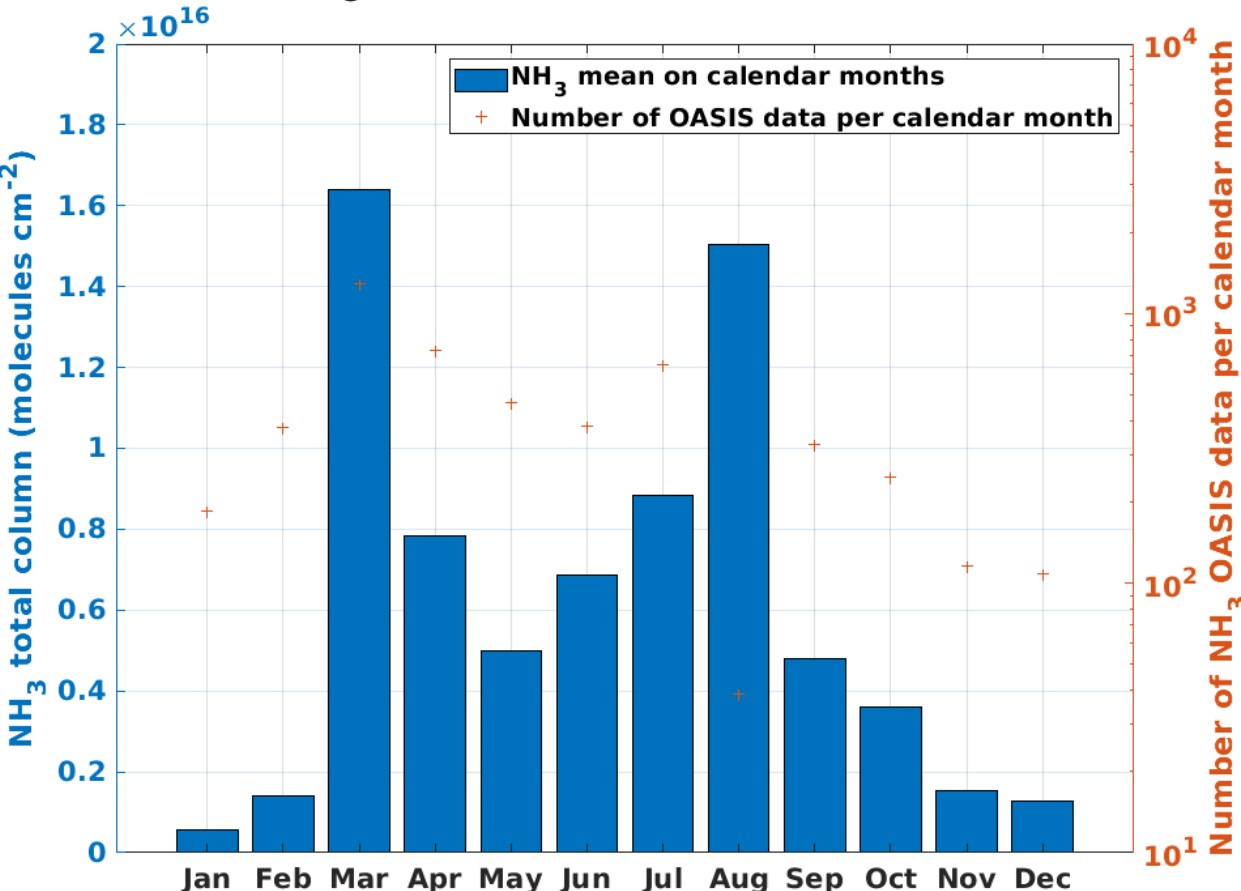

**Figure 9: Average annual cycle of monthly NH₃ total columns (molecules.cm⁻²) observed by OASIS over Paris, averaged over the 2009-2017 period. The red crosses represent the number of OASIS data per calendar month on a log scale (right axis).**

**Figure 10: Correlation of daily NH$_3$ total columns (molecules.cm$^{-2}$) retrieved from OASIS measurements and daily PM2.5 concentrations (µg.m$^{-3}$) derived from in-situ measurements of the Airparif network (Vitry-sur-Seine station), plotted by season.**

10 **For each season, R is the correlation coefficient and N is the number of coincident observations between atmospheric ammonia and PM2.5.**





| NH$_3$ error | NH$_3$ column (molecules.cm$^{-2}$) | | |
|---|---|---|---|
| | average winter NH$_3$ (0.12 $\times 10^{16}$) | average NH$_3$ (0.84 $\times 10^{16}$) | max NH$_3$ (9.1 $\times 10^{16}$) |
| Random error (%) | 23.3 | 6.7 | 1.8 |
| Systematic error (%) | 22.6 | 20.8 | 20.0 |
| Total error (%) | 32.4 | 21.8 | 20.1 |

Table 1: Random and systematic errors according to NH$_3$ total columns. The total errors, combining the systematic and random errors vary from 20% up to 35%. Note that 9.1 10$^{16}$ molecules/cm$^2$ is the maximum ammonia column measured over Paris during the 2009-2017 period (exactly in March 2012).



| Season | No. | Mean ($\times 10^{16}$ molecules.cm$^{-2}$) | Median ($\times 10^{16}$ molecules.cm$^{-2}$) | Standard deviation ($\times 10^{16}$ molecules.cm$^{-2}$) |
|---|---|---|---|---|
| MAM | 2486 | 1.17 | 0.83 | 1.03 |
| JJA | 1070 | 0.84 | 0.73 | 0.44 |
| SON | 691 | 0.38 | 0.31 | 0.28 |
| DJF | 673 | 0.12 | 0.09 | 0.12 |
| all year | 4920 | 0.84 | 0.62 | 0.86 |

**Table 2: Statistics of the retrieved NH$_3$-OASIS measurements depending on the season (MAM for March/April/May, JJA for June/July/August, SON for September/October/November, DJF for December/January/February), and for all the calendar year. (No.: Number of spectra, Mean, Median and Standard deviation). Total columns are given in 1 $\times 10^{16}$ molecules.cm$^{-2}$.**