# Peer review of "Atmospheric ammonia $(NH_3)$ over the Paris megacity: 9 years of total column observations from ground-based infrared remote sensing"

_Atmospheric Measurement Techniques, 2019_

## Referee Comment (RC1) · Anonymous Referee #1 · 3 Oct 2019

Review of Tournadre et al.,

The paper presents an extensive and highly usable data record of FTIR-NH3. Without any doubt it will be very helpful for future air quality evaluation and model and satellite validations. There are not many locations in the world with such an extensive and long term NH3 record, and only a few with instruments with the capability to measure the total column of NH3 at high temporal resolution. The paper is easy to read but could use some restructuring and editing of the text. The sections on the FTIR retrieval and the comparison with IASI are interesting, but section 3.3 seems added on and could be removed without too much impact to the manuscript. For example PM2.5 is barely

mentioned in the introduction. While comparing FTIR-NH3 to pm2.5 is interesting, a more complete analysis and interpretation using a model will be needed if the authors want to keep the section.

Major comments.

1. The retrieval fits are performed over a very wide window. While the authors claim that this is needed, have tests been performed for smaller windows? Past results with the more high resolution FTIR have shown problems with very wide windows, which was one of the reasons to use smaller micro windows (Dammers et al., 2015). The FTIR used by the UNAM team in Mexico City is also a VERTEX and they have reported succesfull fits with smaller windows (Dammers et al., 2017). A comparison of Figure 2 and 3 (maybe merge the figure?) shows that the strongest signatures in the residual correlate well with the location of the strongest NH3 lines. While the fits with an SD of 2% are excellent, compared to the weak absorption feature of NH3 this can still result in a large offset of the NH3 total columns. If possible add a % based fit and take a look at the % deviation around the NH3 lines (maybe mark the locations like in Figure 2). In the text the authors mentioned that HITRAN 2008 was used. Dammers et al 2015 and most of the NDACC FTIR teams used HITRAN 2012 in combination with a few CO2 line adjustments. This can potentially improve the spectral fits.

2. The PROFFIT retrieval seems to be based on a scaling method instead of a full physical retrieval (although I can be mistaken, but as far as I can see it is not mentioned in the text) therefore the choice of the NH3 apriori profile shape is quite essential. The authors mention in section 2.3 (this should be moved into 2.2 probably) that they use a climatological ammonia profile. Does this profile vary monthly? Furthermore, can some more information (or a figure with the shape) be provided on how it compares to profiles used in other studies/products?, for example the profile used in the IASI-NNv2.X product (Van Damme et al., 2017), the CrIS-NH3 product (Shephard and Cady-Pereira, 2015), and the NDACC-FTIR retrievals (Dammers et al., 2015). As mentiond by Van Damme et al. (2014) the choice of profile shape in a column based
retrieval can easily vary the results by a factor 2. A similar result seems to be found by the authors as they mention on P6/line 22-24 with a relative different of +20%. What makes the MIPAS profile optimal in this sense? Did the other tested apriori produce worse fits?

3. The averaging kernel or observational operator are an essential piece of information but are completely missing in the text. The OASIS-NH3 instrument should be superior in its sensitivity to the lower boundary layer compared to satellite measurements. A figure and short discussion of the (total column) averaging kernel can go a long way in helping us understand where the sensitivity of the retrieval lies and why there are differences compared to IASI.

4. This brings us to the comparison of OASIS-NH3 to IASI-NH3. The authors reference the results in Dammers et al., 2015 but that study focussed on an older version of IASI, IASI-LUT. Dammers et al., 2017 reports the results using a more recent version of IASI-NH3, IASI-NNv1 (Figure A1). The slope of S=0.96 for that product is a lot better than the reported S=0.6 for the older product. Van Damme et al., 2018 also state that the most recent version of IASI-NN shows even better results and a lower bias for higher total columns, which would mean we can expect a better comparison. One of the reasons can be found in the absence of the use of an averaging kernel to adjust the IASI total columns to the same playing field. The current comparison can be seen as incomplete as its uncertain where the sensitivity of both instruments lie, and potentially we're comparing the NH3 in the mixing layer to half the boundary layer or the effect of a different apriori (shape).

5. The authors show a initial comparison of OASIS-Nh3 to nearby pm2.5 measurements. While this is interesting it feels somewhat out of place. PM2.5 is barely mentioned in the introduction and only pops up at the end of section 3. Furthermore, most facts are referenced from other studies and the improvement that this study brings, both the high temporal resolution of the FTIR and the vertical total column, are not really used in the analysis. If the authors want to keep the section on PM2.5, an improved

comparison will be needed, with for example the help of a model for interpretation. The in review study by Viatte et al., 2019 for example, shows similar results with a more extensive analysis of the Ile de France region.

6. Something that the author could add instead (but not essential to the text!) is an initial analysis of the diurnal variability, which should not take to long to produce. The authors did excellent work on getting such a long dataseries and have around 5000 measurements spread over 9 years, which accounting for overcast days would mean around 5-10 meaurements a day. Spread out every 15 minutes this must show some diurnal variability of the NH3 total column concentrations (for example split by season) and I for one would be very interested to see that instead of a comparison to PM2.5.

Minor comments and edits.

1. Split section 2 in 2.1 for FTIR, 2.2 with a description of IASI, 2.3 with a description on PM2.5. this will improve the readability and is easier for reference of retrieval characteristics, uncertainties etc. 2. Maybe move section 3.2 up before the comparison with IASI. First completely describe the dataset and variabilities before moving to the comparison with IASI. This can help in the interpretation of any differences between the two. 3. Section 3.1: The authors choose a collocation criteria of 15 km and 30 min while the study that they compare their results with (Dammers et al., 2016) uses 50 km and 90 minutes. Do your results change a lot when using those criteria? Using wider criteria should increase the number of observations, as only 50 measurements out of 5000 initial measurements remain.

Some smaller edits:

1. P2 L21, there have been several studies recently covering the lifetime of NH3. If possible reference Lutsch et al., 2016, Van Damme 2018 and Dammers et al., 2019.

2. P3. L 13: add some examples of networks with high temporal resolution measuremenets (for example LML in the Netherlands, Volten et al., 2013)

3. P3. L20: the correct reference for CrIS would be Shephard and Cady-Pereira 2015. GOSAT also has a Nh3 product: Someya et al., 2019.

4. P3. L28-31, not important for the intro, move to dataset section.

5. P8. L20-21. Although I somewhat agree with the statement, the underestimation can also be caused by other sources. Also the averaging kernel/observational operator has not been applied therefor the results can not be directly compared to the results in Dammers et al., 2016. Explore some further causes of the underestimation (apriori choice) or show some supporting proof that the sensitivity is the cause (which should somewhat be resolved by the use of the averaging kernel).

References:

Dammers, E., Vigouroux, C., Palm, M., Mahieu, E., Warneke, T., Smale, D., Langerock, B., Franco, B., Van Damme, M., Schaap, M., Notholt, J., and Erisman, J. W.: Retrieval of ammonia from ground-based FTIR solar spectra, Atmos. Chem. Phys., 15, 12789–12803, https://doi.org/10.5194/acp-15-12789-2015, 2015.

Dammers, E., Shephard, M. W., Palm, M., Cady-Pereira, K., Capps, S., Lutsch, E., Strong, K., Hannigan, J. W., Ortega, I., Toon, G. C., Stremme, W., Grutter, M., Jones, N., Smale, D., Siemons, J., Hrpcek, K., Tremblay, D., Schaap, M., Notholt, J., and Erisman, J. W.: Validation of the CrIS fast physical NH3 retrieval with ground-based FTIR, Atmos. Meas. Tech., 10, 2645–2667, https://doi.org/10.5194/amt-10-2645-2017, 2017.

Dammers, E., McLinden, C. A., Griffin, D., Shephard, M. W., Van Der Graaf, S., Lutsch, E., Schaap, M., Gainairu-Matz, Y., Fioletov, V., Van Damme, M., Whitburn, S., Clarisse, L., Cady-Pereira, K., Clerbaux, C., Coheur, P. F., and Erisman, J. W.: NH3 emissions from large point sources derived from CrIS and IASI satellite observations, Atmos. Chem. Phys., 19, 12261–12293, https://doi.org/10.5194/acp-19-12261-2019, 2019.

Lutsch, E., Dammers, E., Conway, S., & Strong, K. (2016). Long-range Transport of

NH3, CO, HCN and C2H6 from the 2014 Canadian Wildfires. Geophysical Research Letters, (43), 8286-8297. https://doi.org/10.1002/2016GL070114.

Shephard, M. W. and Cady-Pereira, K. E.: Cross-track Infrared Sounder (CrIS) satellite observations of tropospheric ammonia, Atmos. Meas. Tech., 8, 1323–1336, https://doi.org/10.5194/amt-8-1323-2015, 2015.

Someya, Y., Imasu, R., Shiomi, K., and Saitoh, N.: Atmospheric ammonia retrieval from the TANSO-FTS/GOSAT thermal infrared sounder, Atmos. Meas. Tech. Discuss., https://doi.org/10.5194/amt-2019-49, in review, 2019.

Van Damme, M., Clarisse, L., Heald, C. L., Hurtmans, D., Ngadi,Y., Clerbaux, C., Dolman, A. J., Erisman, J. W., and Coheur,P. F.: Global distributions, time series and error characterizationof atmospheric ammonia (NH3) from IASI satellite observations,Atmos. Chem. Phys., 14, 2905–2922, doi:10.5194/acp-14-2905-2014, 2014.

Van Damme, M., Whitburn, S., Clarisse, L., Clerbaux, C., Hurtmans, D., and Coheur, P.-F.: Version 2 of the IASI NH3 neural network retrieval algorithm: near-real-time and reanalysed datasets, Atmos. Meas. Tech., 10, 4905–4914, https://doi.org/10.5194/amt-10-4905-2017, 2017.

Van Damme, M., Clarisse, L., Whitburn, S., Hadji-Lazaro, J., Hurtmans, D., Clerbaux, C., & Coheur, P. F. (2018). Industrial and agricultural ammonia point sources exposed. Nature, 564(7734), 99.

Viatte, C., Wang, T., Van Damme, M., Dammers, E., Meleux, F., Clarisse, L., Shephard, M. W., Whitburn, S., Coheur, P. F., Cady-Pereira, K. E., and Clerbaux, C.: Atmospheric ammonia variability and link with PM formation: a case study over the Paris area, Atmos. Chem. Phys. Discuss., https://doi.org/10.5194/acp-2019-138, in review, 2019.

---

## Short Comment (SC1) · 12 Oct 2019

A correction has to be applied concerning the approach to select IASI data coinciding with OASIS ground-based:

The sentence "We also exclude NH3-IASI data with absolute errors higher than $5\,10^{-15}$ molecules cm$^{-2}$, or relative errors higher than 100 % (suppressing also very low and negative ammonia columns)."

shall be replaced by:

"We exclude NH3-IASI data with relative errors higher than 100 %, except those with

absolute errors lower than $5 \times 10^{-15}$ molecules cm-2 . IASI data with negative values are also excluded."

---

## Referee Comment (RC2) · Anonymous Referee #2 · 21 Oct 2019

Review of : "Atmospheric ammonia (NH3) over the Paris megacity: 9 years of total column observations from ground-based infrared remote sensing "

In situ data records of NH3, especially over a long period, are very rare. This paper presents a nine year dataset of NH3 total column measurements from an FTIR located in a suburb of Paris. This is a valuable resource for validating NH3 satellite data and for determining the relationships between satellite and surface measurements. That the data is from one of the largest cities in Europe makes it especially relevant, given that the fraction of the world's population living in urban areas is increasing significantly, with all the attendant air quality problems. The paper provides both a mostly very clear

description of the data and the results of a comparison against the IASI ANNI-NH3-v2.2R product, along with analyses of the seasonal variability captured by the FTIR NH3 and correlations between the FTIR data and measured PM2.5 amounts.

I have many of the same comments posted by Reviewer#1, and will refer to them in the next few paragraphs.

Comment 2: The authors need to confirm that the PROFIT retrievals only provides a scaling factor. As the first reviewer stated, this type of retrieval can be strongly influenced by the a priori profile shape; therefore showing a plot of the selected a priori profile and comparing it against the a priori profiles used in the papers listed is an excellent idea. The authors did test a different a priori, but did not state which one and without the plot we suggest it is not possible to ascertain how different it is from the selected a priori.

Comments 3 and 4: here I partially disagree with the first reviewer: if the PROFIT algorithm does not retrieve a profile it cannot provide an averaging kernel (AK); the IASI product also does not generate an AK for each observation, though some AKs are available (see van Damme et al., 2014). The authors could use an optimal estimation algorithm (possibly the FORLI code (van Damme et al., 2014)) on a subset of the data in order to obtain an AK that could provide at least a sense of the vertical sensitivity of the OASIS-NH, then compare it to the IASI AK. I leave it to the editor to decide if this exercise is required. It would certainly be useful for interpreting the results.

Comment 5: The section on PM2.5 is poorly written and not very informative; it should either be expanded and rewritten or eliminated.

Comment 6: Here I strongly agree that a section on diurnal variability observed by OASIS-NH3 would be very interesting and useful, since there are large uncertainties in the diurnal cycle. However, it appears the authors will present this analysis in a separate paper. Can they confirm?

Comment on Figure 7: Can the authors explain why the slopes increase with increasing dmin, until about 120 km, then decrease again?

Minor edits and comments (suggested changes are in bold)

Page 3 Lines 13-16: ...infrared remote sensing from satellites. ... These methods measure over large footprints rather than at points, but are noticeably .... Current space-based NH3 data are available from the IASI ... Line 19: ...Partnership, Shephard and Cady-Pereira, 2015, Dammers ... Line 28: The authors should contact the IASI team for their estimates of the IASI-NH3 precision and uncertainty.

Page 4 Line 9: ...located in the Paris suburbs ... Line 15: Is there any rejection criterion based on weak signals?

Page 6 Line 13: ...spectra, so the degree of freedom for the ammonia retrievals is 1. Line 23: ... differences represented by the error bars ...

Page 7 Line 7: Our analysis is the first compassion of surface NH3 measurements from a megacity with NH3-IASI data and covers seven years of data Line 26: omit colocation criteria, as it has just been cited above.

Page 8 Line 6: ...centered on the OASIS ... Line 8: ... the 15 km width of the rings was chosen to minimize the impact of ammonia spatial variability and to maximize ... Line 11: ...show the number of coincident ... Line 21: ... between the surface and the lower troposphere and to the spatial variability of the IASI footprint.

Page 9 Line 27: ... its impact on the concentrations of fine ...

Page 11 Line 11: Besides lower sensitivity to the surface ammonia concentrations, spatial heterogeneity within the IASI footprint can lead to lower values. Line 13-22: These sentences required some rewriting for clarity; my suggestions are below.

This study used the 9 year OASIS-NH3 time series to focus on seasonal variability of atmospheric NH3 in the Paris region. The predominance of NH3 peaks occurring

in March is particularly noticeable: all measurements above $2*10^{16}$ molecules NH3 cm-2 , which corresponds to the mean of data plus one standard deviation over the springtime period (March/April/May), occur in this month, and are well correlated with manure spreading time periods (Ramanantenasoa et al., 2018).

The sentence below is confusing. It's not clear if mineral fertilizers are applied in spring or summer, and if their application contributes to the March or summer peak.

Mineral fertilizers are mainly applied in Île-de-France region because there are major arable crop (especially cereals) farming areas, which could generate high ammonia concentrations under sunny conditions, when the solar OASIS measurements are performed.

This study also found high summer values above $1.5*10^{16}$ molecules NH3 cm-2 , which corresponds roughly to the mean of data plus one standard deviation over the June/July/August time period, which could be due to increased volatility of ammonia under warm meteorological conditions.

Please also note the supplement to this comment:
https://www.atmos-meas-tech-discuss.net/amt-2019-301/amt-2019-301-RC2-supplement.pdf

---

## Author Comment (AC1) · 31 Jan 2020

Review of Tournadre et al. in AMT

Correspondence to: Pascale Chelin (pascale.chelin@lisa.u-pec.fr)

Atmospheric ammonia (NH3) over the Paris megacity: 9 years of total column observations from ground-based infrared remote sensing

Anonymous Referee #1

Referee: The paper presents an extensive and highly usable data record of FTIR-NH3. Without any doubt it will be very helpful for future air quality evaluation and model and satellite validations. There are not many locations in the world with such an extensive and long term NH3 record, and only a few with instruments with the capability to measure the total column of NH3 at high temporal resolution. The paper is easy to read but could use some restructuring and editing of the text. The sections on the FTIR retrieval and the comparison with IASI are interesting, but section 3.3 seems added on and could be removed without too much impact to the manuscript. For example PM2.5 is barely mentioned in the introduction. While comparing FTIR-NH3 to pm2.5 is interesting, a more complete analysis and interpretation using a model will be needed if the authors want to keep the section.

Authors: First of all, we would like to thank the referee for his/her constructive and useful comments which served us as a guideline for compiling the revised version of the manuscript. All comments are addressed as detailed below. We agree with the restructuration of the paper, editing and withdrawing section 3.3 about particles, as done in the revised manuscript (RM) of the paper.

**Major comments.**

1. The retrieval fits are performed over a very wide window. While the authors claim that this is needed, have tests been performed for smaller windows? Past results with the more high resolution FTIR have shown problems with very wide windows, which was one of the reasons to use smaller micro windows (Dammers et al., 2015). The FTIR used by the UNAM team in Mexico City is also a VERTEX and they have reported succesfull fits with smaller windows (Dammers et al., 2017). A comparison of Figure 2 and 3 (maybe merge the figure?) shows that the strongest signatures in the residual correlate well with the location of the strongest NH3 lines. While the fits with an SD of 2% are excellent, compared to the weak absorption feature of NH3 this can still result in a large offset of

the NH3 total columns. If possible add a % based fit and take a look at the % deviation around the NH3 lines (maybe mark the locations like in Figure 2).

Authors: Clarified. The choice of using large microwindows aims at retrieving gas abundances from the intensity contrast between the target gas signature and the surrounding continuum. If the spectral range for the determination of the continuum is too limited, the ability of correctly reconstructing the target gas amount is likely decreasing, while the result that fit residuals are reduced when the spectral window is restricted to lesser number of points is not necessarily linked to a better quality of the retrieval of the gas abundance. In former times, computational burden was enforcing the use of narrow microwindows in practical work, with the hardware available today this constraint is less relevant. The approach of using narrow microwindows is still used in some cases. More recent fit strategies developed for example for TCCON apply rather broad microwindows for their primary target species (molecular oxygen, carbon dioxide, and methane). This choice can induce problems also if the background continuum over the entire selected spectral window used cannot be modelled down to the spectral noise level for spectroscopic or instrumental reasons (see discussion on this matter in Kiel et al, 2016). Obviously the optimum strategy is to include an empirical smooth background continuum fit, which preserves as much information on the contrast between target spectral lines and surrounding continuum, while still allowing for compensation of background continuum variation beyond our rigorous spectral modelling capability. We decided to maintain the broad microwindows, but the referee's comment induced some further testing on the applied background fit, as a result from this investigation, we decided to fit an empirical background with four degrees of freedom instead of the linear background we used before (the background fitting model used by PROFFIT is approximately equivalent to cubic splines).

In the RM, we didn't merge Figures 2 and 3 because we thought it is not necessary for clarity but we marked the NH3 line locations with arrows close to the fits. We clarify this issue on large microwindows as follows (lines 21-23, page 5) "The choice of using large microwindows aims at retrieving  $NH_3$  abundances from the intensity contrast between the target gas signature and the surrounding continuum. We account for this last one by fitting an empirical background polynomial function with respect to wavelength, with four degrees of freedom."

Kiel, M., D. Wunch, P. O. Wennberg, G. C. Toon, F. Hase, and T. Blumenstock: Improved retrieval of gas abundances from near-infrared solar FTIR spectra measured at the Karlsruhe TCCON station, Atmos. Meas. Tech., 9, 669-682, doi:10.5194/amt-9-669- 2016, 2016

Referee: In the text the authors mentioned that HITRAN 2008 was used. Dammers et al 2015 and most of the NDACC FTIR teams used HITRAN 2012 in combination with a few CO2 line adjustments. This can potentially improve the spectral fits.

Authors: Clarified. We also exchanged the linelists by HITRAN 2012 (which slightly but consistently improves fit quality) but prefer to avoid additional ad-hoc changes on line parameters. This is clarified in the RM as (lines 8-9 page 6) "which does not include ad-hoc changes on line parameters added in HITRAN 2012".

2. The PROFFIT retrieval seems to be based on a scaling method instead of a full physical retrieval (although I can be mistaken, but as far as I can see it is not mentioned in the text) therefore the choice of the NH3 a priori profile shape is quite essential. The authors mention in section 2.3 (this should be moved into 2.2 probably) that they use a climatological ammonia profile. Does this profile vary monthly? Further-more, can some more information (or a figure with the shape) be provided on how it compares to profiles used in other studies/products?, for example the profile used in the IASI-NNv2.X product (Van Damme et al., 2017), the CrIS-NH3 product (Shephard and Cady-Pereira, 2015), and the NDACC-FTIR retrievals (Dammers et al., 2015).

Authors: Clarified. We confirm that a scaling factor retrieval was used (in Section 2.2) for all the NH3-OASIS time series and mention it in the text (line 15, page 10 in the new Section 3.3). We also added information about the MIPAS a priori profile in Section 2.3 (line 19 page 6) of the revised paper.

As mentioned by Van Damme et al. (2014) the choice of profile shape in a column based retrieval can easily vary the results by a factor 2. A similar result seems to be found by the authors as they mention on P6/line 22-24 with a relative different of +20%. What makes the MIPAS profile optimal in this sense? Did the other tested a priori produce worse fits? Authors: A different a priori profile, that is close to those used in NDACC-FTIR retrievals for Bremen and Lauder, was also tested and described in Section 2.3 of the revised paper. While it reduces the spectral fits by 60%, it shows very similar temporal variability with relative differences which are of the same order of magnitude than the ammonia total column error (only +20% relative difference for ammonia total columns retrieved with the new a priori). In that case, results did not vary by a factor of 2.

This is clarified in the RM as (lines 29-33 page 6) "Using this a priori profile reduces the mean squared difference between measured and simulated spectra by about 60%. However, both retrievals with homogenous and sloped a priori profiles show rather similar results, with the same relative evolution in time and differences in absolute terms in the order of magnitude of the total column retrieval error (the use of the sloped a priori profile increases the retrieved NH3 abundances by 20% with respect to that using an homogenous a priori)".

3. The averaging kernel or observational operator are an essential piece of information but are completely missing in the text. The OASIS-NH3 instrument should be superior in its sensitivity to the lower boundary layer compared to satellite measurements. A figure and short discussion of the (total column) averaging kernel can go a long way in helping us understand where the sensitivity of the retrieval lies and why there are differences compared to IASI.

Authors: Agreed. as required by both referees, a paragraph (new section 3.3) and additional Figure 10 about NH3-OASIS sensitivity were added in the revised paper (lines 14-30, page 10): "3.3 Vertical distribution of sensitivity of the NH3-OASIS approach

As mentioned in section 2.2, the NH3-OASIS dataset presented in Figs. 4, 8 and 9 is derived from a scaling factor retrieval scheme whose state vector only has one scalar

value associated with the NH3 abundance. Therefore, this approach does not provide an averaging kernel matrix as optimal estimation or Tikhonov schemes do, but only a single value of degrees-of-freedom (DOF) without any information on the vertical distribution of the retrieval sensitivity. In order to estimate the vertical sensitivity to NH3 provided by OASIS measurements, we have performed a few tests using a NH3 profile retrieval scheme applied to OASIS spectra with a Tikhonov-Phillips regularization (as similarly implemented for ozone profiles by Viatte et al., 2011). Figure 10 presents examples of averaging kernel diagonals for NH3 profile retrievals based on OASIS spectra measured on 13 March 2014, at different times of the day and thus different solar zenith angles (SZA). We remark that OASIS measurements may provide information on the abundance of NH3 located around 500 m, with maximum sensitivity for smaller solar zenith angles corresponding to thicker air masses (occurring in the early morning or late afternoon). These OASIS averaging kernel diagonals peak at similar altitudes as those estimated by Dammers et al. (2017) for a high spectral resolution Fourier Thermal Infrared spectrometer at the Pasadena site (peaking around 940 hPa, thus approximately at 600 m above sea level). These altitudes are typically located within the atmospheric boundary layer during springtime and summer, at mid-latitudes where most of the atmospheric NH3 column variability is expected to occur. Additional tests (not shown) using different spectroscopic databases (HITRAN 2008 and HITRAN 2012) change very little the estimation of the sensitivity of the OASIS retrieval."

4. This brings us to the comparison of OASIS-NH3 to IASI-NH3. The authors reference the results in Dammers et al., 2015 but that study focussed on an older version of IASI, IASI-LUT. Dammers et al., 2017 reports the results using a more recent version of IASI-NH3, IASI-NNv1 (Figure A1). The slope of S=0.96 for that product is a lot better than the reported S=0.6 for the older product. Van Damme et al., 2018 also state that the most recent version of IASI-NN shows even better results and a lower bias for higher total columns, which would mean we can expect a better comparison. One of the reasons can be found in the absence of the use of an averaging kernel to adjust the IASI total columns to the same playing field. The current comparison can be seen as incomplete as its uncertain where the sensitivity of both instruments lie, and potentially we're comparing the NH3 in the mixing layer to half the boundary layer or the effect of a different a priori (shape).

Authors: Clarified. The differences in the sensitivity to NH3 between OASIS and IASI retrievals are clarified in the manuscript. We believe that these differences explain the underestimation of IASI with respect to OASIS, and therefore in a slope of 0.73 to 0.8. This is clarified as (lines 7-9 page 10) "This underestimation may be explained by IASI's lower sensitivity to surface ammonia concentrations, due to the coarse spectral resolution and weak thermal contrast between the surface and the lower troposphere and to the spatial heterogeneity of ammonia within the IASI footprint".

The results of the comparison between the FTIR at Bremen and the new IASI data is mention in the RM as (lines 16-18 page 9) "and NH3-IASI neural network version: R = 0.67 and slope of 0.96 for 802 coincidences from several ground-based FTIR stations (Dammers et al., 2017)".

5. The authors show a initial comparison of OASIS-Nh3 to nearby pm2.5 measurements. While this is interesting it feels somewhat out of place. PM2.5 is barely mentioned in the introduction and only pops up at the end of section 3. Furthermore, most facts are referenced from other studies and the improvement that this study brings, both the high temporal resolution of the FTIR and the vertical total column, are not really used in the analysis. If the authors want to keep the section on PM2.5, an improved

comparison will be needed, with for example the help of a model for interpretation. Thein review study by Viatte et al., 2019 for example, shows similar results with a more extensive analysis of the lle de France region.

Authors: Agreed. in the revised version, the section 3.3 on PM2.5 has been withdrawn, as NH3 diurnal variation observed by OASIS and its impact on ammonium particles will be analyzed in details in a future separate paper. Only two sentences in the conclusions are kept in order to inform the readers (lines 31-33 page 11 and lines 1-7 page 12) "Since ammonia is a major precursor of PM2.5 over Europe, as shown by e.g. Fortems-Cheiney et al. (2016) during a European spring haze episode, we expect a link between high ammonia concentrations and inorganic salts, such as ammonium nitrate. That period during late 2012 winter (documented by Petit et al. (2014)), was probably the most polluted month of March of the last ten years in Paris region (Petit et al., 2017) with the highest NH3-OASIS total columns in the period 2009-2017 over the Paris region. The link between ammonia concentrations and the formation and volatilization of fine particles such as ammonium salts is beyond the scope of this paper and will be discussed in a future study on the diurnal analysis of total and surface ammonia measurements from Paris region during a high spring pollution event.". Also, the previous Figure 10 was suppressed.

6. Something that the author could add instead (but not essential to the text!) is an initial analysis of the diurnal variability, which should not take too long to produce. The authors did excellent work on getting such a long data series and have around 5000 measurements spread over 9 years, which accounting for overcast days would mean around 5-10 measurements a day. Spread out every 15 minutes this must show some diurnal variability of the NH3 total column concentrations (for example split by season) and I for one would be very interested to see that instead of a comparison to PM2.5.

Authors: Clarified. We confirm, as asked by Referee #2, that analysis of the NH3 diurnal variation which can be observed by OASIS using a long data series with measurements spread out every 10 minutes in case of continuous sunny conditions, is dedicated to a next separate paper, during spring pollution events over Paris region. See the citation in previous comment.

**Minor comments and edits.**

1. Split section 2 in 2.1 for FTIR, 2.2 with a description of IASI, 2.3 with a description on PM2.5. this will improve the readability and is easier for reference of retrieval characteristics, uncertainties etc.

Authors: Clarified. as section 3.3 on PM2.5 has been withdrawn, we decided not to change section 2 and only describe in details the ground-based remote-sensing measurements.

2. Maybe move section 3.2 up before the comparison with IASI. First completely describe the dataset and variabilities before moving to the comparison with IASI. This can help in the interpretation of any differences between the two.

Authors: Agreed. As suggested by referee #1, section 3.2 moved up before the comparison with IASI helping to interpret any differences between the two remotesensing data series. Numbers of Figures have consequently been modified.

3. Section 3.1: The authors choose a collocation criteria of 15 km and 30 min while the study that they compare their results with (Dammers et al., 2016) uses 50 km and 90 minutes. Do your results change a lot when using those criteria? Using wider criteria should increase the number of observations, as only 50 measurements out of 5000 initial measurements remain.

Authors: Clarified. We tested a co-location criteria of 50 km and 90 minutes as proposed by Referee #1. We observed that, using these wider criteria, the number of observations is obviously increasing but not so different correlation is found. This is clarified in the RM as (lines 10-11 page 9): "Tests with wider coincidence criteria (50 km and +/- 90 minutes) do not show significant differences (similar correlations are obtained despite a greater number of coincidences)."

**Some smaller edits:**

1. P2 L21, there have been several studies recently covering the lifetime of NH3. If possible reference Lutsch et al., 2016, Van Damme 2018 and Dammers et al., 2019. Authors: Agreed. More recent studies covering the lifetime of NH3 have been included as mentioned.

2. P3. L 13: add some examples of networks with high temporal resolution measurements (for example LML in the Netherlands, Volten et al., 2013). Authors: Agreed. The established EMEP and LML networks have been included in the revised paper with the correspondent references.

3. P3. L20: the correct reference for CrIS would be Shephard and Cady-Pereira 2015.GOSAT also has a Nh3 product: Someya et al., 2019. Authors: Corrected and completed. The correct reference for CrIS (also mentioned by Referee 2) has been added in the revised version and also the GOSAT reference.

4. P3. L28-31, not important for the intro, move to dataset section. Authors: Agreed. As suggested, lines concerning the precision of NH3-IASI data are moved to dataset section 3.2.

5. P8. L20-21. Although I somewhat agree with the statement, the underestimation can also be caused by other sources. Also the averaging kernel/observational operator has not been applied therefore the results can not be directly compared to the results in Dammers et al., 2016. Explore some further causes of the underestimation (a priori choice) or show some supporting proof that the sensitivity is the cause (which should somewhat be resolved by the use of the averaging kernel).

Authors: Agreed and clarified. As detailed in the answers above, we have performed an analysis of the sensitivity of the  $NH_3$ -OASIS retrieval (new section 3.3) and also added as possible reason for the underestimation of IASI with respect to OASIS the heterogeneity of  $NH_3$  within the IASI footprint. On the other hand, the IASI approach is based on neural networks and it does not provide averaging kernels nor uses a priori profiles that could be compared with those from OASIS.

References:

Dammers, E., Vigouroux, C., Palm, M., Mahieu, E., Warneke, T., Smale, D., Langerock, B., Franco, B., Van Damme, M., Schaap, M., Notholt, J., and Erisman, J. W.: Retrievalof ammonia from ground-based FTIR solar spectra, Atmos. Chem. Phys., 15, 12789–12803, https://doi.org/10.5194/acp-15-12789-2015, 2015.

Dammers, E., Shephard, M. W., Palm, M., Cady-Pereira, K., Capps, S., Lutsch, E.,Strong, K., Hannigan, J. W., Ortega, I., Toon, G. C., Stremme, W., Grutter, M., Jones, N., Smale, D., Siemons, J., Hrpcek, K., Tremblay, D., Schaap, M., Notholt, J., andErisman, J. W.: Validation of the CrIS fast physical NH3 retrieval with ground-basedFTIR, Atmos. Meas. Tech., 10, 2645–2667, https://doi.org/10.5194/amt-10-2645-2017,2017.

Dammers, E., McLinden, C. A., Griffin, D., Shephard, M. W., Van Der Graaf, S., Lutsch, E., Schaap, M., Gainairu-Matz, Y., Fioletov, V., Van Damme, M., Whitburn, S., Clarisse, L., Cady-Pereira, K., Clerbaux, C., Coheur, P. F., and Erisman, J. W.: NH3 emissionsfrom large point sources derived from CrIS and IASI satellite observations, Atmos.Chem. Phys., 19, 12261–12293, https://doi.org/10.5194/acp-19-12261-2019, 2019.

Lutsch, E., Dammers, E., Conway, S., & Strong, K. (2016). Long-range Transport of C5NH3, CO, HCN and C2H6 from the 2014 Canadian Wildfires. Geophysical ResearchLetters, (43), 8286-8297. https://doi.org/10.1002/2016GL070114.

Shephard, M. W. and Cady-Pereira, K. E.: Cross-track Infrared Sounder (CrIS) satel-lite observations of tropospheric ammonia, Atmos. Meas. Tech., 8, 1323–1336,https://doi.org/10.5194/amt-8-1323-2015, 2015.

Someya, Y., Imasu, R., Shiomi, K., and Saitoh, N.: Atmospheric ammonia retrievalfrom the TANSO-FTS/GOSAT thermal infrared sounder, Atmos. Meas. Tech. Discuss., https://doi.org/10.5194/amt-2019-49, in review, 2019.

Van Damme, M., Clarisse, L., Heald, C. L., Hurtmans, D., Ngadi,Y., Clerbaux, C., Dolman, A. J., Erisman, J. W., and Coheur,P. F.: Global distributions, time series and errorcharacterizationof atmospheric ammonia (NH3) from IASI satellite observations,Atmos.Chem. Phys., 14, 2905–2922, doi:10.5194/acp-14-2905-2014, 2014.

Van Damme, M., Whitburn, S., Clarisse, L., Clerbaux, C., Hurtmans, D., and Coheur, P.-F.: Version 2 of the IASI NH3 neural network retrieval algorithm: near-real-time and reanalysed datasets, Atmos. Meas. Tech., 10, 4905–4914, https://doi.org/10.5194/amt-10-4905-2017, 2017.

Van Damme, M., Clarisse, L., Whitburn, S., Hadji-Lazaro, J., Hurtmans, D., Clerbaux, C., & Coheur, P. F. (2018). Industrial and agricultural ammonia point sources exposed.Nature, 564(7734), 99.

Viatte, C., Wang, T., Van Damme, M., Dammers, E., Meleux, F., Clarisse, L., Shephard, M. W., Whitburn, S., Coheur, P. F., Cady-Pereira, K. E., and Clerbaux, C.: Atmosphericammonia variability and link with PM formation: a case study over the Paris area, Atmos. Chem. Phys. Discuss., https://doi.org/10.5194/acp-2019-138, in review, 2019.

---

## Author Comment (AC2) · 31 Jan 2020

Review of Tournadre et al. in AMT

Correspondence to: Pascale Chelin (pascale.chelin@lisa.u-pec.fr)

Atmospheric ammonia (NH3) over the Paris megacity: 9 years of total column observations from ground-based infrared remote sensing

Anonymous Referee #2

Referee: In situ data records of NH3, especially over a long period, are very rare. This paper presents a nine year dataset of NH3 total column measurements from an FTIR located in a suburb of Paris. This is a valuable resource for validating NH3 satellite data and for determining the relationships between satellite and surface measurements. That the data is from one of the largest cities in Europe makes it especially relevant, given that the fraction of the world's population living in urban areas is increasing significantly, with all the attendant air quality problems. The paper provides both a mostly very clear description of the data and the results of a comparison against the IASI ANNI-NH3-v2.2R product, along with analyses of the seasonal variability captured by the FTIRNH3 and correlations between the FTIR data and measured PM2.5 amounts.

Authors: First of all, we would like to thank the referee for his/her constructive and useful comments which served us as a guideline for compiling the revised version of the manuscript. All comments are addressed as detailed below. We agree with the restructuration of the paper, editing and withdrawing section 3.3 about particles, as done in the revised manuscript (RM) of the paper.

Referee: I have many of the same comments posted by Reviewer#1, and will refer to them in the next few paragraphs.

Comment 2: The authors need to confirm that the PROFIT retrievals only provides a scaling factor.

Authors: Clarified. We confirm that a scaling factor retrieval is used (in Section 2.2) for all the NH3-OASIS time series and state it clearly in the text (in Section 2.2, lines 11-12 page 6) "A scaling factor of a climatological vertical profile of NH3 is adjusted in order to minimize the difference between measured and simulated spectra, so the degree of

freedom for the ammonia retrievals is 1". We also added information about the MIPAS a priori profile in Section 2.3 of the revised paper (lines 19-20 page 6) "A climatological a priori profile of NH3 that assumes fixed vertically homogenous NH3 concentrations (0.1 ppb) in the troposphere is taken from the MIPAS project (Remedios et al., 2007).".

As the first reviewer stated, this type of retrieval can be strongly influenced by the a priori profile shape; therefore showing a plot of the selected a priori profile and comparing it against the a priori profiles used in the papers listed is an excellent idea. The authors did test a different a priori, but did not state which one and without the plot we suggest it is not possible to ascertain how different it is from the selected a priori.

Authors: Clarified. A different a priori profile, that is closed to those used in NDACC-FTIR retrievals for Bremen and Lauder, was also tested and described in Section 2.3 of the revised paper (lines 27-29 page 6) "Another a priori was tested with higher concentrations of NH3 in the atmospheric boundary layer (with fixed concentrations up to 900 m then decreasing NH3 concentrations until 4 km).". While it reduces the spectral fit residual by 60%, it shows the same variabilities with relative differences which are of the same order of magnitude than the ammonia total column error (only +20% relative difference for ammonia total columns retrieved with the new a priori). In that case, results did not vary by a factor of 2.

Comments 3 and 4: here I partially disagree with the first reviewer: if the PROFFIT algorithm does not retrieve a profile it cannot provide an averaging kernel (AK); the IASI product also does not generate an AK for each observation, though some AKs are available (see van Damme et al., 2014). The authors could use an optimal estimation algorithm (possibly the FORLI code (van Damme et al., 2014)) on a subset of the data in order to obtain an AK that could provide at least a sense of the vertical sensitivity of the OASIS-NH3, then compare it to the IASI AK. I leave it to the editor to decide if this exercise is required. It would certainly be useful for interpreting the results.

Authors: Agreed and test performed. In order to estimate the full averaging kernels matrix, we have implemented a profile retrieval approach and tested it on a subset of the data. This corresponds to the new section 3.3 and the new Figure 10 (lines 14-30, page 10): "3.3 Vertical distribution of sensitivity of the NH3-OASIS approach

As mentioned in section 2.2, the NH3-OASIS dataset presented in Figs. 4, 8 and 9 is derived from a scaling factor retrieval scheme whose state vector only has one scalar value associated with the NH3 abundance. Therefore, this approach does not provide an averaging kernel matrix as optimal estimation or Tikhonov schemes do, but only a single value of degrees-of-freedom (DOF) without any information on the vertical distribution of the retrieval sensitivity. In order to estimate the vertical sensitivity to NH3 provided by OASIS measurements, we have performed a few tests using a NH3 profile retrieval scheme applied to OASIS spectra with a Tikhonov-Phillips regularization (as similarly implemented for ozone profiles by Viatte et al., 2011). Figure 10 presents examples of averaging kernel diagonals for NH3 profile retrievals based on OASIS spectra measured on 13 March 2014, at different times of the day and thus different solar zenith angles

(SZA). We remark that OASIS measurements may provide information on the abundance of NH3 located around 500 m, with maximum sensitivity for smaller solar zenith angles corresponding to thicker air masses (occurring in the early morning or late afternoon). These OASIS averaging kernel diagonals peak at similar altitudes as those estimated by Dammers et al. (2017) for a high spectral resolution Fourier Thermal Infrared spectrometer at the Pasadena site (peaking around 940 hPa, thus approximately at 600 m above sea level). These altitudes are typically located within the atmospheric boundary layer during springtime and summer, at mid-latitudes where most of the atmospheric $NH_3$ column variability is expected to occur. Additional tests (not shown) using different spectroscopic databases (HITRAN 2008 and HITRAN 2012) change very little the estimation of the sensitivity of the OASIS retrieval."

Comment 5: The section on PM2.5 is poorly written and not very informative; it should either be expanded and rewritten or eliminated.

Authors: Agreed and withdrawn. In the revised version, the section 3.3 on PM2.5 has been eliminated because NH3 diurnal variation observed by OASIS and its impact on ammonium particles will be analyzed in details in a next separate paper. Only two sentences in the conclusions are kept in order to inform the readers (lines 31-33 page 11 and lines 1-7 page 12) "Since ammonia is a major precursor of PM2.5 over Europe, as shown by e.g. Fortems-Cheiney et al. (2016) during a European spring haze episode, we expect a link between high ammonia concentrations and inorganic salts, such as ammonium nitrate. That period during late 2012 winter (documented by Petit et al. (2014)), was probably the most polluted month of March of the last ten years in Paris region (Petit et al., 2017) with the highest NH3-OASIS total columns in the period 2009-2017 over the Paris region. The link between ammonia concentrations and the formation and volatilization of fine particles such as ammonium salts is beyond the scope of this paper and will be discussed in a future study on the diurnal analysis of total and surface ammonia measurements from Paris region during a high spring pollution event." As a consequence, the previous Figure 10 was suppressed.

Comment 6: Here I strongly agree that a section on diurnal variability observed by OASIS-NH3 would be very interesting and useful, since there are large uncertainties in the diurnal cycle. However, it appears the authors will present this analysis in a separate paper. Can they confirm?

Authors: Clarified. We can confirm that analysis of the NH3 diurnal variation which can be observed by OASIS using a long data series with measurements spread out every 10 minutes in case of continuous sunny conditions, is dedicated to a next separate paper, during spring pollution events over Paris region.

Comment on Figure 7: Can the authors explain why the slopes increase with increasing dmin, until about 120 km, then decrease again?

Authors: Clarified. We can assume that until about 120 km, we include, step by step, areas close to agricultural regions such as Picardie (Amiens city) and Champagne (Reims city) with higher NH3-IASI amounts as seen in the map: Global ammonia point sources as seen by IASI satellite instruments, provided by L. Clarisse and M. Van Damme in https://www2.ulb.ac.be/cpm/NH3-IASI.html (Clarisse et al., 2019; Van Damme et al., 2018). Beyond 120 km, we would expect more horizontal heterogeneity of the NH3 sources and abundances and therefore the slope decreases accordingly. This is clarified in the RM (lines 9-12 page 10) as "The regression slope increases until 120 km for dmin and decreases beyond. This might be linked to the fact that the main surrounding agricultural regions (e.g. Picardie and Champagne) are located until about 120 km away from OASIS, and therefore NH3 sources (Clarisse et al., 2019; Van Damme et al., 2018), and these sources are more heterogenous beyond this distance.".

**Minor edits and comments** (suggested changes are in bold)

Referee: Page 3 Lines 13-16:...infrared remote sensing from satellites.... These methods measure over large footprints rather than at points, but are noticeably....

Authors: Clarified. The statement is meant for remote sensing techniques in general, not only from space.

Current space-based NH3 data are available from the IASI...

Authors: Corrected. The suggested change has been included.

Referee: Line 19:...Partnership, Shephard and Cady-Pereira, 2015, Dammers...

Authors: Reference added. This paper has been added in page 3 as proposed, and in the references.

Line 28:
The authors should contact the IASI team for their estimates of the IASI-NH3 precision and uncertainty.

Authors: Agreed and done. So based on NH3-IASI data from the ANNI-NH3-v2.2R baseline version, used for the comparison, we calculated average and median errors which are 89% and 60%, respectively. These values are now given in Section 3.2 of the revised paper (lines 24-28 page 8) "Average and median errors of these satellite measurements used in this study are 89% and 60% respectively, which are coherent with uncertainties for most of the $NH_3$-IASI data listed in Van Damme et al. (2014 and 2017), because of small absorption features by ammonia observed with the relatively coarse spectral resolution of IASI as compared to ground-based instruments.

Referee: Page 4 Line 9: . . . located in the Paris suburbs...

Authors: Corrected. The suggested change has been included.

Line 15: Is there any rejection criterion based on weak signals ?

Authors: Clarified. There is no criterion based on weak signals, but on signal-to-noise ration lower than 30, (lines 12-14 page 6) "As the radiance values are rather small below 1000 cm$^{-1}$, a quality criterion was introduced selecting only spectra with a signal-to-noise ratio higher than 30"

Page 6 Line 13: . . . spectra, so the degree of freedom for the ammonia retrievals is 1. Authors: Corrected. The suggested change has been included.

Referee: Line 23: . . . differences represented by the error bars...

Authors: Corrected. We have added another expression to clarify our argument.

Referee: Page 7 Line 7: Our analysis is the first comparison of surface NH3 measurements from a megacity with NH3-IASI data and covers seven years of data.

Authors: Added. The suggested sentence has been included.

Referee: Line 26: omit colocation criteria, as it has just been cited above.

Authors: Corrected. The colocation criteria have been suppressed.

Referee: Page 8 Line 6: . . . centered on the OASIS . . .

Authors: Corrected. The suggested change has been included.

Referee: Line 8: . . . the 15 km width of the rings was chosen to minimize the impact of ammonia spatial variability and to maximize . . .

Authors: Corrected. The suggested change has been included.

Referee: Line 11: . . . show the number of coincident . . .

Authors: Corrected. The suggested change has been included.

Referee: Line 21: . . . between the surface and the lower troposphere and to the spatial variability of the IASI footprint.

Authors: Corrected. The suggested change has been included.

Referee: Page 9 Line 27: . . . its impact on the concentrations of fine . . .

Authors: Corrected. The sentence has been modified as follows (lines 2-3, page 12) "The link between ammonia concentrations and the formation and volatilization of fine particles such as ammonium salts ".

Referee: Page 11 Line 11: Besides lower sensitivity to the surface ammonia concentrations, spatial heterogeneity within the IASI footprint can lead to lower values. Authors: Corrected. The suggested change has been included.

Referee: Line 13-22: These sentences required some rewriting for clarity; my suggestions are below.

This study used the 9 year OASIS-NH3 time series to focus on seasonal variability of atmospheric NH3 in the Paris region. The predominance of NH3 peaks occurring in March is particularly noticeable: all measurements above $2*10^{16}$ molecules NH3cm-2 , which corresponds to the mean of data plus one standard deviation over the springtime period (March/April/May), occur in this month, and are well correlated with manure spreading time periods (Ramanantenasoa et al., 2018).The sentence below is confusing. It's not clear if mineral fertilizers are applied in spring or summer, and if their application contributes to the March or summer peak. Mineral fertilizers are mainly applied in Île-de-France region because there are major arable crop (especially cereals) farming areas, which could generate high ammonia concentrations under sunny conditions, when the solar OASIS measurements are performed. This study also found high summer values above $1.5*10^{16}$ molecules NH3 cm-2 ,which corresponds roughly to the mean of data plus one standard deviation over the June/July/August time period, which could be due to increased volatility of ammonia under warm meteorological conditions.

Authors: Corrected. The sentences, that needed rewriting for clarity, have been included as suggested.

---

## Author Comment (AC3) · 10 May 2020

Review of Tournadre et al. in AMT

Correspondence to: Pascale Chelin (pascale.chelin@lisa.u-pec.fr)

Atmospheric ammonia (NH3) over the Paris megacity: 9 years of total column observations from ground-based infrared remote sensing

Anonymous Referee #1

Referee: As I mentioned in my previous review, the paper presents an extensive and highly usable data record of FTIR-NH3. Without any doubt it will be very helpful for future air quality evaluation and model and satellite validations. There are not many locations in the world with such an extensive and long term NH3 record, and only a few with instruments with the capability to measure the total column of NH3 at high temporal resolution. The revised paper shows a lot of improvement and I only have a few minor comments and suggestions.

Authors: First of all, we would like to thank the referee for his/her constructive and useful comments which served us as a guideline for compiling and improving the second revised version of the manuscript. All minor comments and edits are addressed as detailed below.

**Minor comments.**
1. Page 6, line 16, as you mention the fits are pretty good with a stdev of 2%. The green arrows point out the main absorption features of ammonia. These however also show that at those line positions the fit is the worst. While partially a result of the lower resolution, a similar thing was seen in Dammers et al., 2015, Fig.3. Potentially add a small discussion point about the uncertainty in the line parameters. The authors already sort of remark this on Page 11 Line 9.

Authors: Clarified.
The PROFFIT retrieval algorithm is a well-recognized method within the NDACC community and the present work on NH$_3$ is based on the HITRAN 2008 spectroscopic database. For the intensities of the relevant lines of the $\nu_2$ band used in our retrieval, an uncertainty index of 4 is generally given (i.e. between 10% and 20%). A paper by Down et al. 2013 compares several experimental sources for NH$_3$ intensities in the region 770-1200 cm$^{-1}$ with a consistency better than ~4%. The best theoretical total band intensity for the $\nu_2$ band is reported as $2.26 \times 10^{-17}$ cm$^{-1}$/(molecule cm$^{-2}$) as compared to the total band intensity of $2.21 \times 10^{-17}$ cm$^{-1}$/(molecule cm$^{-2}$) in HITRAN 2014, i.e. a difference of 2 % consistent with the uncertainty quoted previously. This systematic uncertainty is certainly contributing to a small fraction of the retrieval uncertainty on the NH$_3$ column given in the

present paper. The residuals of the fit as seen in Figure 2 and Figure 3 are smaller than 2 % and some residuals do present some correlation with $NH_3$ lines. However, the effect is within the stated uncertainty of the retrieved column amount. Several effects are intermixed in the simultaneous retrieval of $NH_3$ together with $CO_2$ and $H_2O$, the lines of which have to be fitted properly. Uncertainty on air-broadening coefficients of $NH_3$ and their temperature dependence and also the line-mixing effects should have also a contribution. So the residuals shown in the figures are to be considered as typical and can vary from one fit to the other.

To have a compromise about these complex question concerning spectroscopy, we propose these updated sentences (Page 6, Lines 31-34 and Page 7, Line 1) in the second revised version. "The total errors are dominated by the combination of uncertainties in the spectroscopic parameters (including also the interfering species), the noise in the spectra and the hypotheses on the retrieval (*a priori* profile of $NH_3$, forward model uncertainties). They are comparable to those estimated by Dammers et al. (2015) for a high resolution ground-based station at Bremen (Germany). In complement to these, according to a review paper on $NH_3$ spectroscopic parameters (Down et al., 2013) the uncertainty of 20% on line intensities is probably a worst case estimate."

2. Page 6, Line 29-33. If I understand correctly the fit improves by 60%! (Or do the authors mean that the fit mismatch increase by 60 %?) When using the sloped apriori profiles, and shows an increase in the NH3 abundances. What is the reason for not switching to these aprioris? Even though the seasonal/temporal patterns do not change, this offset/ratio will change the comparison with IASI.

Authors: Clarified. This a priori profile is just one possibility and is empirically chosen. We are currently developing a new version of the OASIS-$NH_3$ retrieval algorithm. Among the different changes, we will change the a priori profile in order to reduce both the spectral fit and the retrieval error (as compared to IASI data for example). We are currently testing several a priori profiles, issued by chemistry-transport models for example, in order to find the best one. These changes will lead to a new version of the OASIS dataset that will be presented in a future publication.

The following sentence in the current version of the paper mentions this coming work (Page 12 Lines 20-21) "Tests of different a priori $NH_3$ profiles will also be performed for reducing the spectral residuals between measured and simulated spectra while providing accurate retrievals of $NH_3$ abundances."

3. Page 8, Line 5-6, maybe remove the "possible" as it's well-known that temperature increases volatility and in many cases seems the reason for higher NH3 temperatures in summer.

Authors: We prefer to keep the adjective "possible" because we would like to be careful with ammonia volatility since not only temperature can impact its abundance, such as the soil properties (soil pH,...) or the properties of the fertilizer applied (organic manure or mineral fertilizer)…

4. Page 9, line 11-12. Excluding negative values will bias the IASI "mean" high, especially in the lower range of the concentrations where the sensitivity is a strong factor. As it's a statistical retrieval, negative values are somewhat to be expected when sensitivity is low, and should average out towards zero (in case of no ammonia).

Authors: Clarified. We understand the fact that IASI data are derived from a statistical approach providing in some cases negative values. However, we chose to exclude all

values with too high uncertainty (>100%) and without physical meaning such as negative ones. We prefer to avoid highly uncertain retrievals. The large majority (98%) of $NH_3$-IASI retrieval corresponding to small $NH_3$ abundances are screened by the retrieval error filter. Therefore, excluding negative values has a negligible effect on the comparison of IASI and OASIS datasets.

This aspect is clarified in the revised manuscript (Page 9 Lines 22-24) as "The majority of IASI retrievals corresponding to weak $NH_3$ abundances are screened out by the relative error criterion, thus the exclusion of negative values has a negligible effect in the comparisons.".

5. Page 9, Lines 19-24. What are the matching mean total columns and relative differences? Most of the values (Fig 6.) seem to be in the range of 0-1.5x1016 molecules cm-2. So 0.78 would be around 50%. Considering IASI's and the FTIR products uncertainty that's about the range we expect for this lower range of total columns. Add some discussion on how these values compare to the uncertainties of the products. Similarly add a short reflection to the conclusions/perspectives.

Authors: Clarified. We think that there is a misunderstood about the range of mean bias between NH3-IASI and NH3-OASI. Indeed in the last version, we mentioned that "The average of the absolute differences is -0.78 $10^{15}$ molecules cm$^{-2}$, with a root mean squared error (RMSE) equal to 4.86 $10^{15}$ molecules cm$^{-2}$ and a standard deviation of error (STDE) equal to 4.84 $10^{15}$ molecules cm$^{-2}$". So mean bias is -0.08 $10^{16}$ molecules cm$^{-2}$ which represents 10% of the average of the values (Fig 6.) in the range of 0-1.5 $10^{16}$ molecules cm$^{-2}$ and 4 time less than $NH_3$-OASIS total retrieval errors of about 20% to 35%. Concerning the matching mean total columns: the average of NH3-IASI is 0.70 $10^{16}$ molecules. cm$^{-2}$ and that for NH3-OASIS is 0.78 $10^{16}$ molecules.cm$^{-2}$.

So, to avoid the confusion and for consistency, we put each $NH_3$ total column in $10^{16}$ molecules.cm$^{-2}$ in the second revised version of the manuscript and added the following statement (Page 10 Lines 2-4) "The mean absolute difference is a factor 10 smaller than the average values of the $NH_3$ abundances (0.78 $10^{16}$ and 0.70 $10^{16}$ molecules cm$^{-2}$ respectively for OASIS and IASI) and also smaller than OASIS total retrieval errors (20 to 35 %)."

6. Page 12, line 6-7. The manuscript already describes the use/test of a different (sloped) a-priori. Shortly reflect on those results?

Authors: Agreed. We have added the additional sentence in the second revised version of the manuscript to mention those tests (Page 12 Lines 21-23) "First tests with a different a priori profile for $NH_3$ show a significant reduction of the residuals between the radiance spectra measured by OASIS and those simulated by PROFFIT."

**Minor edits**
1. Page 2, Line 5, remove "potentially" as its well established what the sources are.
Authors: Agreed. The adverb "potentially" has been suppressed.

2. Page 2, line 6, add a reference for the sources.

Authors: Agreed. Four references already cited in the text have been included related to ammonia sources and sinks.
(Galloway et al., 2003; Behera et al., 2013; Chang et al., 2016; Van Damme et al., 2018).

3. Page 2, Line 21, potentially add a reference to the recent paper by Dammers et al which includes lifetime estimates for NH3 point sources.
Authors: Added. The suggested reference has been included.

4. Page 2, line 26, add a reference for the 50% PM2.5 statement.
Authors : Added. In order to state that ammonium salts can represent 50% of PM2.5, we have added (Page 2 Line 27) the three following references (Bressi et al., 2013; Petit et al., 2014; 2015).
So we have added two of them in the bibliography:
Bressi, M., Sciare, J., Ghersi, V., Bonnaire, N., Nicolas, J. B., Petit, J.-E., Moukhtar, S., Rosso, A., Mihalopoulos, N., and Féron, A.: A one-year comprehensive chemical characterisation of fine aerosol ($PM_{2.5}$) at urban, suburban and rural background sites in the region of Paris (France), Atmos. Chem. Phys., 13, 7825-7844, https://doi.org/10.5194/acp-13-7825-2013, 2013.
Petit, J.-E., Favez, O., Sciare, J., Crenn, V., Sarda-Estève, R., Bonnaire, N., Močnik, G., Dupont, J.-C., Haeffelin, M., and Leoz-Garziandia, E.: Two years of near real-time chemical composition of submicron aerosols in the region of Paris using an Aerosol Chemical Speciation Monitor (ACSM) and a multi-wavelength Aethalometer, Atmos. Chem. Phys., 15, 2985-3005, https://doi.org/10.5194/acp-15-2985-2015, 2015.

5. Page 2, line 26-31, while it's interesting to point out uncertainties in the emissions in the Paris regions, the paper no longer discusses the PM2.5 concentrations, and this part can potentially be shorted or removed from the manuscript.
Authors: Agreed. In the second revised version, as required by the referee, the last sentence on PM2.5 has been withdrawn. We kept the other sentences to highlight the need to have accurate ammonia measurements for improving the air quality models.

6. Page 3, line 19: "current or until very recent space-based NH3 data… etc. add a short line as an introduction how the satellites measure (also Infrared based), and link to the first to show that it's possible: Beer et al., 2008.
Authors: Added. A suggested introduction has been included as follows (Page 3 Lines 21-22) with a link to Beer et al. paper ″A few observations were first reported from space using an advanced IR sounder (Beer et al., 2008) which enables retrievals of atmospheric $NH_3$″. Then we have added this reference in the bibliography.

7. Page 4, Line 2, add some references of studies that used FTIR to validate satellite measurements.

Authors: Added. Different references have been included concerning validation of satellite products with ground-based IR instruments for different trace species (Griesfeller et al., 2006; Strong et al., 2008; Buchholz et al., 2017) and also recently for ammonia (Dammers et al., 2016, 2017).

8. Page 4, Line 20-21: The part about section 3.3 needs to be rewritten as it still point to the previous version of the paper.

Authors: Agreed. We apologize for this error in the introduction with the former section 3.3. A consistent sentence with section 3.3 has been added in the second revised paper (Page 4 Lines 26-27) "Finally, in Section 3.3, a few tests are performed using a $NH_3$ profile retrieval scheme applied to OASIS spectra with a Tikhonov-Phillips regularization, showing the vertical sensitivity to $NH_3$ provided by OASIS measurements. "

9. Page 5, Line 27, mention the green arrows in the figure.

Authors: Added. We have mentioned in comma (Page 5 Line 32) that the strong spectral signatures of ammonia are pointed out by green arrows in Figures 2 and 3 "The strong spectral signatures of ammonia (pointed out by green arrows in Figures 2 and 3). We also added one sentence about green arrows in each caption of Figures 2 and 3.

10. Page 11 Line 2, add "a" between "FTIR instruments" and "moderate spectral resolution".

Authors: Added. The suggested change has been included.

References

Dammers, E., McLinden, C. A., Griffin, D., Shephard, M. W., Van Der Graaf, S., Lutsch, E., Schaap, M., Gainairu-Matz, Y., Fioletov, V., Van Damme, M., Whitburn, S., Clarisse, L., Cady-Pereira, K., Clerbaux, C., Coheur, P. F., and Erisman, J. W.: NH3 emissions from large point sources derived from CrIS and IASI satellite observations, Atmos. Chem. Phys., 19, 12261–12293, https://doi.org/10.5194/acp-19-12261-2019, 2019.

Beer, R., Shephard M. W., Kulawik, S. S., Clough, S. A., Elder-ing, A., Bowman, K. W., Sander, S. P., Fisher, B. M., Payne, V.H., Luo, M., Osterman, G. B., and Worden, J. R.: First satel-lite observations of lower tropospheric ammonia and methanol, Geophys. Res. Lett., 35, L09801,doi:10.1029/2008GL033642,2008.

---

## Referee Report (RR1)

As I mentioned in my previous review, the paper presents an extensive and highly usable data record of FTIR-NH3. Without any doubt it will be very helpful for future air quality evaluation and model and satellite validations. There are not many locations in the world with such an extensive and long term NH3 record, and only a few with instruments with the capability to measure the total column of NH3 at high temporal resolution. The revised paper shows a lot of improvement and I only have a few minor comments and suggestions.

**Minor comments**

- Page 6, line 16, as you mention the fits are pretty good with a stdev of 2%. The green arrows point out the main absorption features of ammonia. These however also show that at those line positions the fit is the worst. While partially a result of the lower resolution, a similar thing was seen in Dammers et al., 2015, Fig.3. Potentially add a small discussion point about the uncertainty in the line parameters. The authors already sort of remark this on Page 11 Line 9.
- Page 6, Line 29-33. If I understand correctly the fit improves by 60%! (Or do the authors mean that the fit mismatch increase by 60 %?) When using the sloped apriori profiles, and shows an increase in the NH3 abundances. What is the reason for not switching to these aprioris? Even though the seasonal/temporal patterns do not change, this offset/ratio will change the comparison with IASI.
- 3. Page 8, Line 5-6, maybe remove the "possible" as it's well-known that temperature increases volatility and in many cases seems the reason for higher NH3 temperatures in summer.
- 4. Page 9, line 11-12. Excluding negative values will bias the IASI "mean" high, especially in the lower range of the concentrations where the sensitivity is a strong factor. As it's a statistical retrieval, negative values are somewhat to be expected when sensitivity is low, and should average out towards zero (in case of no ammonia).
- Page 9, Lines 19-24. What are the matching mean total columns and relative differences? Most of the values (Fig 6.) seem to be in the range of 0-1.5x1016 molecules cm-2. So 0.78 would be around 50%. Considering IASI's and the FTIR products uncertainty that's about the range we expect for this lower range of total columns. Add some discussion on how these values compare to the uncertainties of the products. Similarly add a short reflection to the conclusions/perspectives.
- Page 12, line 6-7. The manuscript already describes the use/test of a different (sloped) a-priori. Shortly reflect on those results?

**Minor edits**

- 1. Page 2, Line 5, remove "potentially" as its well established what the sources are.
- 2. Page 2, line 6, add a reference for the sources.
- 3. Page 2, Line 21, potentially add a reference to the recent paper by Dammers et al which includes lifetime estimates for NH3 point sources.
- 4. Page 2, line 26, add a reference for the 50% PM2.5 statement.
- 5. Page 2, line 26-31, while it's interesting to point out uncertainties in the emissions in the Paris regions, the paper no longer discusses the PM2.5 concentrations, and this part can potentially be shorted or removed from the manuscript.
- Page 3, line 19: "current or until very recent space-based NH3 data... etc. add a short line as an introduction how the satellites measure (also Infrared based), and link to the first to show that it's possible: Beer et al., 2008.
- 7. Page 4, Line 2, add some references of studies that used FTIR to validate satellite measurements.
- 8. Page 4, Line 20-21: The part about section 3.3 needs to be rewritten as it still point to the previous version of the paper.
- 9. Page 5, Line 27, mention the green arrows in the figure.

10. Page 11 Line 2, add "a" between "FTIR instruments" and "moderate spectral resolution". References

Dammers, E., McLinden, C. A., Griffin, D., Shephard, M. W., Van Der Graaf, S., Lutsch, E., Schaap, M., Gainairu-Matz, Y., Fioletov, V., Van Damme, M., Whitburn, S., Clarisse, L., Cady-Pereira, K., Clerbaux, C., Coheur, P. F., and Erisman, J. W.: NH3 emissions from large point sources derived from CrIS and IASI satellite observations, Atmos. Chem. Phys., 19, 12261–12293, https://doi.org/10.5194/acp-19-12261-2019, 2019.

Beer, R., Shephard M. W., Kulawik, S. S., Clough, S. A., Elder-ing, A., Bowman, K. W., Sander, S. P., Fisher, B. M., Payne, V.H., Luo, M., Osterman, G. B., and Worden, J. R.: First satel-lite observations of lower tropospheric ammonia and methanol, Geophys. Res. Lett., 35, L09801, doi:10.1029/2008GL033642, 2008.